# Near-surface magma flow instability drives cyclic lava fountaining at Fagradalsfjall, Iceland

Samuel Scott [1] ✉, Melissa Pfeffer [2], Clive Oppenheimer[3], Enikö Bali [1], Oliver D. Lamb[4,5], Talfan Barnie[2], Andrew W. Woods[6], Rikey Kjartansdóttir[1] & Andri Stefánsson[1]

Lava fountains are a common manifestation of basaltic volcanism. While magma degassing plays a clear key role in their generation, the controls on their duration and intermittency are only partially understood, not least due to the challenges of measuring the most abundant gases, $H_2O$ and $CO_2$. The 2021 Fagradalsfjall eruption in Iceland included a six-week episode of uncommonly periodic lava fountaining, featuring ~100–400 m high fountains lasting a few minutes followed by repose intervals of comparable duration. Exceptional conditions on 5 May 2021 permitted close-range (~300 m), highly time-resolved (every ~2 s) spectroscopic measurement of emitted gases during 16 fountain-repose cycles. The observed proportions of major and minor gas molecular species (including $H_2O$, $CO_2$, $SO_2$, HCl, HF and CO) reveal a stage of $CO_2$ degassing in the upper crust during magma ascent, followed by further gas-liquid separation at very shallow depths (~100 m). We explain the pulsatory lava fountaining as the result of pressure cycles within a shallow magma-filled cavity. The degassing at Fagradalsfjall and our explanatory model throw light on the wide spectrum of terrestrial lava fountaining and the subsurface cavities associated with basaltic vents.

Lava fountains drive a column of gas and liquid or solid clasts extending from a few tens of meters to up to 1 km or more[1–4]. Distinct from 'explosive' and 'effusive' eruption styles, which reflect magma fragmentation at depth in the conduit or at the surface, respectively, lava fountaining involves surface fragmentation under a condition of choked-flow, a behavior typical of rapidly ascending, low viscosity, volatile-poor (<1 wt % $H_2O$) basaltic magmas[5–7]. Fountaining activity is often episodic, as observed at Kīlauea[8] and Etna[9], and in spectacular fashion at Fagradalsfjall on the Reykjanes peninsula in 2021[10,11]. More generally, sustained episodic eruptive activity and/or degassing behavior is observed at many volcanoes, spanning a wide range of eruptive styles and magma rheologies[12–14]. The physical mechanisms

driving episodic behavior remain debated[5], with some models[3–6] emphasizing the role of magma ascent rate and others[15,16] emphasizing gas accumulation in subsurface cavities.

The 2021 eruption at Fagradalsfjall provided an exceptional opportunity to study episodic lava fountaining. In contrast to previous well-studied eruptions of Kīlauea[3–5,8,15] and Etna[9,16], both of which were fed by upper crustal (<5 km depth) magma reservoirs, the eruption at Fagradalsfjall was sourced from a magma reservoir at the mantle/crust interface from which melt was transported to the near-surface without stalling for an extended period (<1 year) in mid- or upper-crustal storage zones[17,18]. Compared with Kīlauea or Etna, the fountaining events at Fagradalsfjall were extremely

[1]Institute of Earth Sciences, University of Iceland, Sturlugata 7, Reykjavík 102, Iceland. [2]Icelandic Meteorological Office, Bústaðavegur 7-9, Reykjavík 105, Iceland. [3]Department of Geography, University of Cambridge, Downing Place, Cambridge CB2 3EN, UK. [4]Department of Earth, Marine and Environmental Sciences, University of North Carolina at Chapel Hill, 104 South Road, Chapel Hill, NC 27599-3315, USA. [5]Te Pū Ao | GNS Science, Wairakei Research Centre, 114 Karetoto Road, RD4, Taupō 3384, New Zealand. [6]BP Institute, University of Cambridge, Cambridge CB3 0EZ, UK. ✉e-mail: samuels@hi.is

regular, with much shorter intervals of fountaining and repose (minutes rather than days/weeks) persisting for more than a month[11]. Noting that comparable basaltic magmas were involved in each case[3,8,18], these observations beg the question: what controls the timescales of fountaining and repose?

We investigate here a high-resolution time series of gas emission compositions (measured using open-path infrared absorption spectroscopy[19]) spanning 16 fountain/repose cycles, along with observations of fountain heights, total gas fluxes and magma geochemistry. In contrast to similar gas measurements performed at Etna[16], which were obtained at a distance of ~1 km from the vent, the optical path for our measurements was only ~300 m. This enabled precise corrections to be made for ambient air contributions to the spectra, yielding well-resolved measurements of volcanic $H_2O$ and $CO_2$ emissions. Given the pressure sensitivity of $H_2O$ solubility in silicate melt as it approaches atmospheric pressure[20,21], and the contrasting

behavior of $CO_2$, which exsolves at greater depth[22], the measured $H_2O$/$CO_2$ ratio reflects the depth and pressure conditions of gas-melt equilibration. By constraining the physical conditions experienced by the gas erupted during fountaining, we probe the process generating intermittency.

## Results and discussion
### Volcanic gas chemistry
Open-path Fourier Transform Infrared (OP-FTIR) observations (see Methods) made on 5 May 2021 ~300 m from vent 5 (Fig. S1, the focus of activity at the time) are reported in Fig. 1. The 140 min of observations captured 16 cycles of alternating fountaining (shaded domains in Fig. 1) and repose spanning two hours and 20 min. Lava fountains reached heights up to ~100 m (black trace in Fig. 1a). During fountaining, the infrared (IR) intensity (blue trace in Fig. 1a) spikes due to the radiance of incandescent pyroclasts. During repose, a lava lake roiled within the

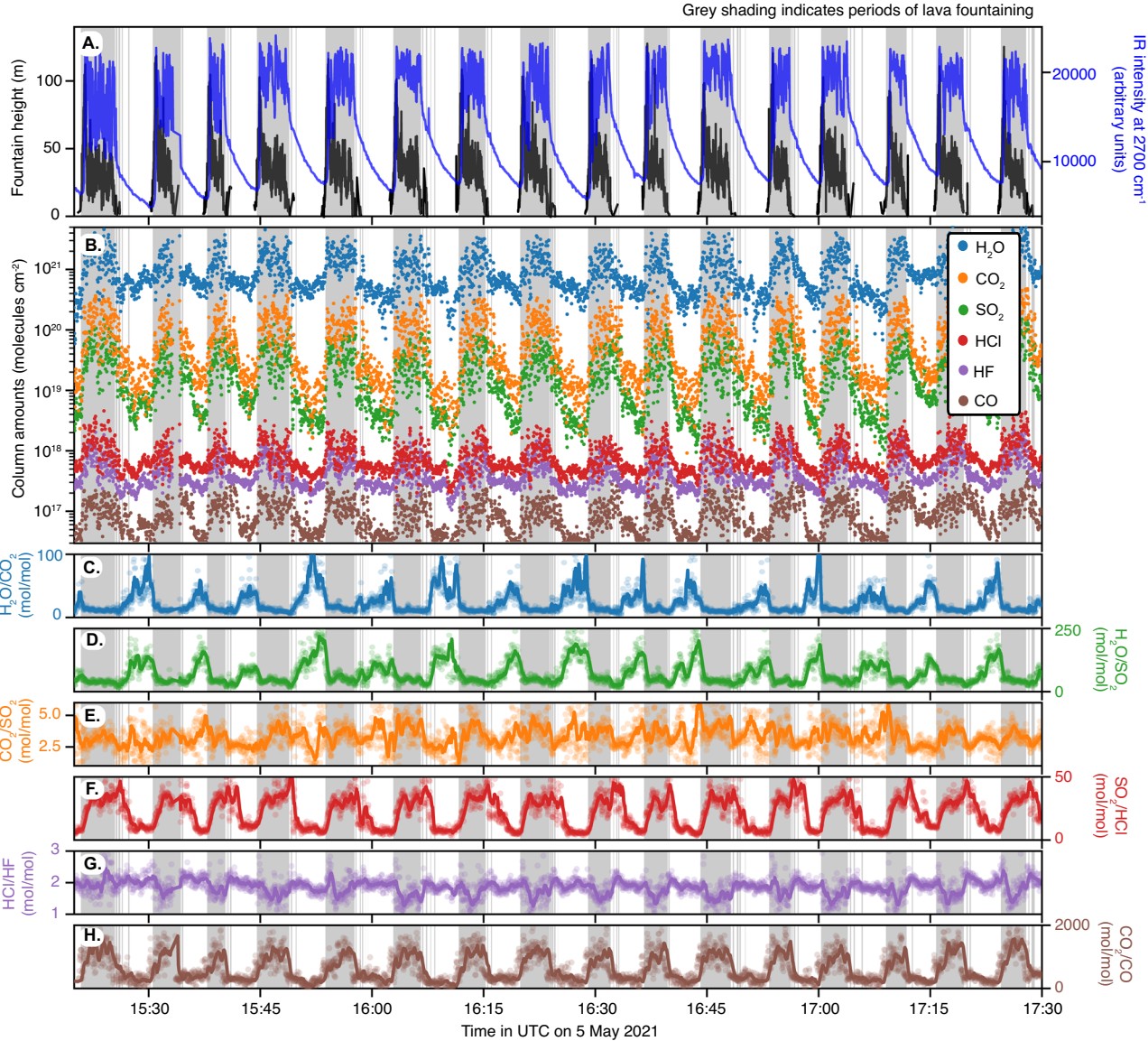

**Fig. 1 | Observations made during of intermittent fountaining of the Fagradalsfjall volcano on 5 May 2021.** Fountaining intervals are delineated by acoustic records (see Methods) shown in gray. **A** Infrared intensity at 2700 cm⁻¹ (blue) and tracked fountain heights (black). **B** Temporal variation in retrieved column amounts (the integrated amount of gas in the instrument's field of view per unit area) for $H_2O$ (blue points), $CO_2$ (orange), $SO_2$ (green), HCl (red), HF (purple), and CO (brown); $H_2O$, $CO_2$ and CO are corrected for atmospheric contributions (see "Methods"). **C–H** Molecular ratios of (**C**). $H_2O$/$CO_2$ (blue), (**D**) $H_2O$/$SO_2$ (green), (**E**) $CO_2$/$SO_2$ (orange), (**F**) $SO_2$/HCl (red), (**G**) HCl/HF (purple), (**H**) $CO_2$/CO (brown). Dots show individual measurements and the solid line the 30 s moving average. Note that all data in (**C–H**) are plotted using a linear scale.

crater beneath its rim. The IR source was now the cooling lava plastered on the inner back wall of the crater.

In the course of an episode, there is a roughly order of magnitude increase in column amounts (reported in molecules per unit area along the optical path) of $H_2O$, $CO_2$, $SO_2$, HCl, HF and CO (Fig. 1b), reflecting increased gas flux. The gas measured during both fountaining and repose is water-rich (>85 mol% $H_2O$) but higher proportions of $CO_2$ and $SO_2$ are recorded for fountains (Table 1). These contrasts in gas composition are emphasized in time-series of molecular ratios (Fig. 1c–h): $H_2O/CO_2$ and $H_2O/SO_2$ decrease sharply from repose to fountaining, from $42 \pm 22$ to $12 \pm 3$ and $121 \pm 41$ to $43 \pm 11$, respectively. While $SO_2/HCl$ increases from $8.7 \pm 3$ during repose to $33 \pm 7$ during fountaining (Fig. 1f), HCl/HF slightly decreases from $1.9 \pm 0.2$ to $1.7 \pm 0.3$ (Fig. 1g). $CO_2/CO$ increases from $310 \pm 130$ in repose to $1101 \pm 371$ during fountaining, with the highest ratios (~2000) measured towards the end of fountaining (Fig. 1h).

**Table 1 | Averaged molecular compositions (±one standard deviation) for fountain and repose gas on 5 May 2021 (15:20:44–17:43:07 UTC)**

| Species abundance | Fountaining $n = 1374$ (mol %) | Repose $n = 1111$ (mol %) |
|---|---|---|
| $H_2O$ | 89.8 ± 1.9 | 96.1 ± 1.5 |
| $CO_2$ | 7.8 ± 1.5 | 2.8 ± 1.3 |
| $SO_2$ | 2.3 ± 0.6 | 0.9 ± 0.3 |
| HCl | 0.08 ± 0.07 | 0.17 ± 0.07 |
| HF | 0.04 ± 0.009 | 0.06 ± 0.03 |
| CO | 0.006 ± 0.004 | 0.010 ± 0.005 |
| Species molecular ratios | (mol/mol) | (mol/mol) |
| $H_2O/CO_2$ | 12.1 ± 3.0 | 41.9 ± 21.6 |
| $H_2O/SO_2$ | 42.7 ± 11.1 | 120.5 ± 41.0 |
| $CO_2/SO_2$ | 3.6 ± 0.9 | 3.35 ± 1.5 |
| $SO_2/HCl$ | 33.0 ± 6.9 | 8.65 ± 2.9 |
| HCl/HF | 1.7 ± 0.3 | 1.90 ± 0.2 |
| $CO_2/CO$ | 1101 ± 371 | 310.4 ± 130 |

$CO_2/SO_2$ changes little during a cycle from $3.6 \pm 0.9$ during fountaining to $3.4 \pm 1.5$ in repose (Fig. 1e).

## Magma degassing from mantle to surface

We modeled the degassing behavior of the Fagradalsfjall melt using D-Compress[23] (see "Methods"), with constraints on initial melt composition and volatile contents from measured compositions of erupted tephra and groundmass glass[18] (see "Methods"). Figure 2a shows the modeled evolution of volatile contents in the Fagradalsfjall melt as it ascends in the crust. As is typical for basaltic magmas[20–22, 24], $CO_2$ begins to exsolve at high pressures (~500 MPa, ~20 km depth), followed by $SO_2$ at intermediate pressure (~100 MPa, ~4 km depth), and $H_2O$ and HCl at low pressures (<20 MPa, <1 km depth). Closed-system degassing of the melt from ~500 MPa to the surface results in $H_2O/CO_2$ (molar) of order 1 (thin dashed line in Fig. 2b), whereas the measured $H_2O/CO_2$ is of order 10 (gray area in Fig. 2b). This discrepancy suggests loss of deeply-exsolved (almost pure $CO_2$) gas at pressures of 20–100 MPa (depth of ~0.5–4 km; Fig. 2a), possibly as a result of bubble growth reaching a critical point favouring separation and escape from the melt[22,25–27]. Such early degassing of $CO_2$ is consistent with pre-eruption uplift and seismicity, interpreted as a signal of magmatic $CO_2$ ingress into nearby geothermal systems at 4 km depth[28]. Assuming closed-system degassing during magma ascent followed by a first stage of gas loss at 20–100 MPa, we can reset the dissolved volatile contents of the melt accordingly then model a second stage of closed- and open-system degassing of the now partially-degassed melt to the surface (Fig. 2b–d). This yields much closer correspondence between the modeled and measured gas compositions.

Figure 2b–d shows that the measured $H_2O/CO_2$, $H_2O/SO_2$ and $CO_2/SO_2$ ratios during fountaining (gray shading in Fig. 2) are all consistent with low pressure equilibration. In particular, the measured $H_2O/CO_2$ of $12 \pm 3$ during fountaining suggests equilibration at near-atmospheric pressures for the closed-system degassing scenario, and pressure of ~2 MPa for an open-system degassing scenario. Although the magmastatic depths corresponding to these pressure estimates depend on assumed magma density and vesicularity, these results suggest gas-melt equilibration at depths of <100 m, i.e., very near to the surface. As gas bubble expansion is rapid in the near-surface, causing the dynamic overpressure that drives the fountaining[30], we

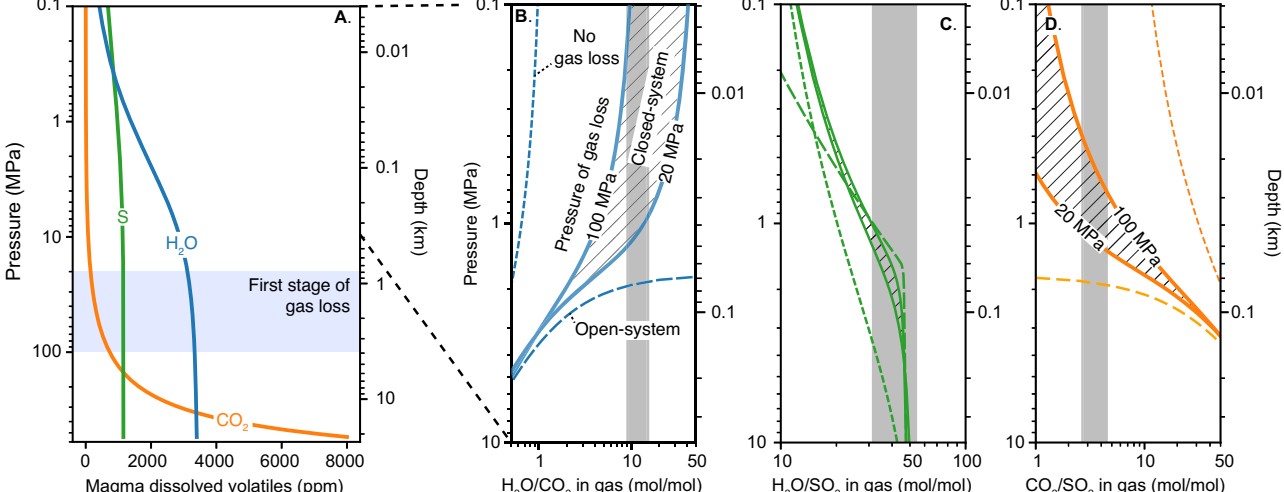

**Fig. 2 | Degassing of the Fagradalsfjall melt. A** Dissolved $CO_2$ (orange line), $H_2O$ (blue), and S (green) concentrations in the melt as a function of pressure. The blue area shows the inferred pressure at which a first stage of gas loss occurs. **B** $H_2O/CO_2$ (blue lines), and (**C**) $H_2O/SO_2$ (green), and (**D**) $CO_2/SO_2$ (orange). Closed-system degassing of initial melt with no gas loss shown with thin dashed line. Closed- and open-system degassing after gas loss at 20–100 MPa shown with solid and thick-dashed lines, respectively, labeled with the assumed pressure of gas loss. Measured ratios during fountaining (plus one standard deviation) shown with gray bars. Inferred pressure of gas equilibration shown in red. Inferred depths are shown on the secondary y-axes assuming a magma density of 2600 kg m⁻³ (ref. 29) and a vesicularity of 0%.

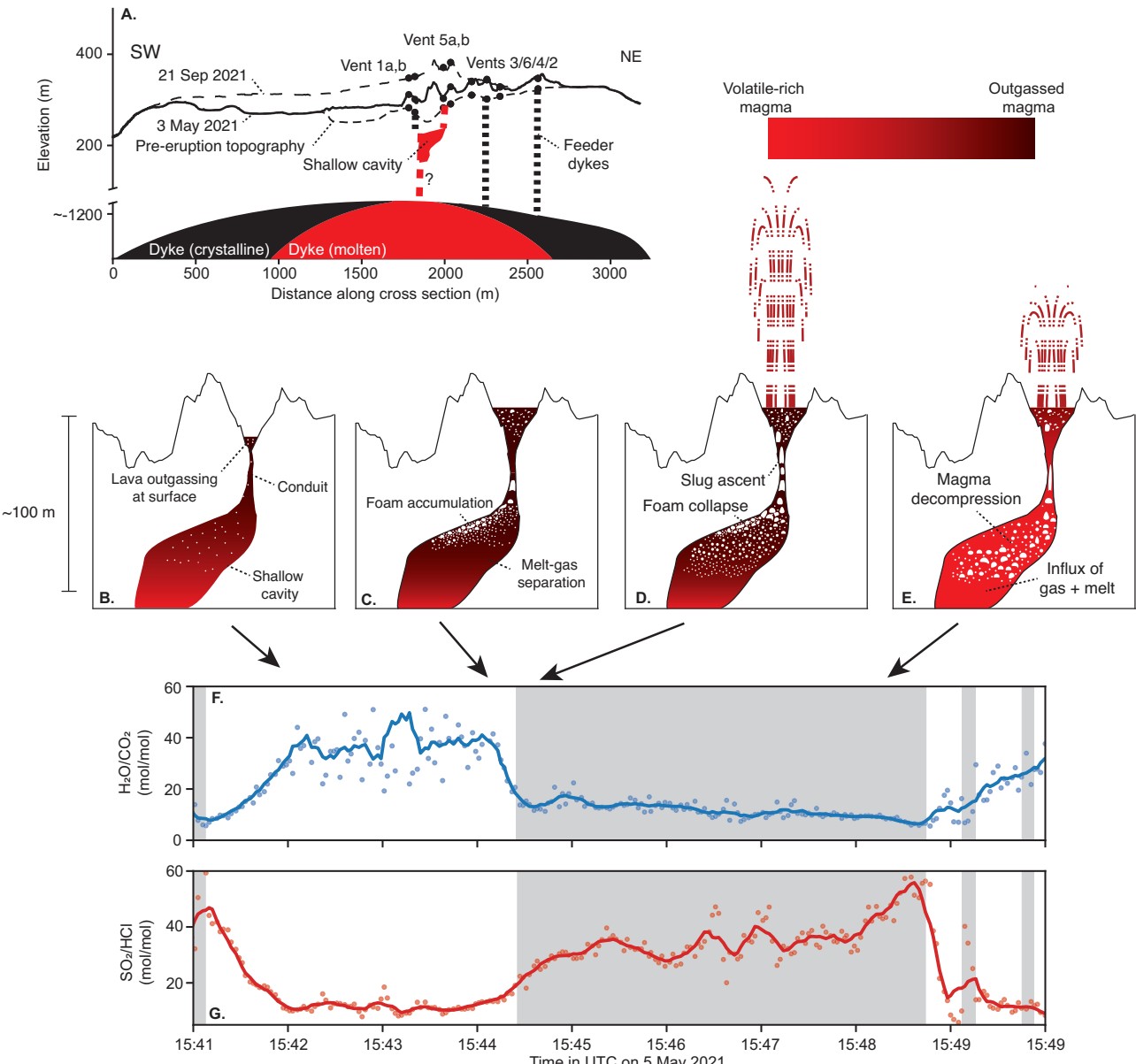

**Fig. 3 | Sketch of the lava fountaining cycle at Fagradalsfjall. A** Three-dimensional perspective of eruption site, with vent 5 (the locus of eruption activity at the time, yellow star) fed by a melt conduit connected to a partially-solidified dyke at -1.5 km depth[17]. Topography[51] along the axis of the dyke shown for pre-eruption conditions, 3 May 2021 and 21 Sep 2021. The cross-sections in (**B**–**E**) focus in on an area where a conduit. The color indicates the degree of outgassing of magma, with red colors indicating 'fresh' magma and black indicating outgassed magma. **B** At the beginning of repose, the crater is filled with outgassed lava, and the level of the lava "lake" decreases as pressure at the base of the conduit equili-brates with the pressure in the underlying cavity. **C** Melt-gas separation in the cavity causes bubbles accumulation on the roof of the cavity to form a foam. Some gas

bubbles leak through the conduit to the surface, decreasing the average density of the magma in the conduit/lake and increasing the level of the lava lake. **D** Ascent of a large slug through the conduit destabilizes the foam layer, which empties from the cavity into the conduit, leading to slug coalescence and tall fountains (-100 m). **E** Lower pressures at the base of the conduit during fountaining promote gas exsolution in the shallow cavity and influx of volatile-rich melt from depth. The progressive degassing of magma in the shallow cavity drives shorter fountains (-30 m) with a gradual waning in the intensity of the fountaining until dense, out-gassed melt drains back into the conduit. The lower panels show the spectroscopic measurements for one of these fountaining cycles (see Fig. 1), relating (**F**) $H_2O/CO_2$ (blue) and (**G**) $SO_2/HCl$ (red) to the fountaining cycle.

therefore suggest that these measurements during fountaining pre-serve an approach to equilibrium attained at shallow depth. However, it should be noted that diffusion-limited $CO_2$ degassing could also contribute to elevated $H_2O/CO_2$ ratios observed during fountaining[31].

The higher $H_2O/CO_2$ and $H_2O/SO_2$ ratios and lower $SO_2/HCl$ ratios measured during repose are characteristic of outgassing[32,33] of residual lava in the lake and uppermost region of the conduit that has already lost most of its initial volatile load during lava fountaining. The mea-sured $CO_2$, $H_2O$, and S concentrations in groundmass glass are 18 ± 3.9, 720 ± 63, and 186 ± 100 ppm, respectively[18], whereas equilibrium

degassing models predict residual $CO_2$, $H_2O$, and S concentrations in magma at atmospheric pressure of <1, ~400, and ~700 ppm, respec-tively (Fig. 3b, c). Thus, the models do not capture the last stages of magma decompression and degassing as pressure approaches atmo-spheric, and non-equilibrium (e.g., diffusional) effects likely come into play[31,34].

## Physical mechanism of fountaining
Several contrasting conceptual models have been invoked to explain the type of episodic fountaining observed at Fagradalsfjall and other

volcanoes. The rise speed dependent (RSD) model[3–5] posits that Hawaiian-style lava fountaining results from rapid magma ascent and coupled magma-gas flow through the conduit, while more periodic Strombolian eruptions result from ascent of decoupled gas slugs through a column of magma. On the other hand, the collapsing foam (CF) model[15, 35, 36] considers a bubble-rich pocket accumulated at the roof of a body of magma at depth via bubble-melt separation. Following collapse of the foam into the conduit due to exceedance of a critical stability threshold, the gas rises and expands, generating the dynamic overpressure that drives the fountain. Another proposed mechanism for episodic activity is tied to dynamic pressure instabilities in lava lake/conduit systems[37,38]. Ascending gas slugs drive magma from the conduit into the lake, gradually filling the lake with magma and increasing the pressure at the base of the conduit, thereby reducing the flow of melt and gas into the lake. When a large slug drains through the conduit, the resulting void refills with denser outgassed magma from the lake. If the pressure at the base of the conduit exceeds that of the source reservoir, the flow at the base of the conduit reverses, resulting in partial drainage of the lake.

In addition to the obvious regularity of the fountaining cycles, observations of the active vent in early May 2021 at Fagradalsfjall (vent 5) indicate that the level and intensity of roiling (degassing) in the "lava lake" in the upward-flaring conduit/vent system increased in the final ~30-60 s of repose (Supplementary Materials, Movie S1). Fountain heights would then increase up to the maximum height (~100–130 m) over ~20 s, prior to an extended period (~3-4 min) of sustained low level fountaining fluctuating at 20–50 m. These fluctuations in fountain height and intensity were mirrored by a transition from more acoustic-tremor-dominated seismic energy during the initial fountaining stage to more Strombolian-style activity with distinct high-amplitude impulsive waveforms[10]. The transition to repose would often be gradual, occurring over ~60 s with short bursts of activity at the end of fountaining, and then gentle spattering of the magma would persist throughout repose, with many small bubbles (~10 cm diameter) and occasional larger bubbles up to ~1 m in diameter. During repose, the upper surface of the lava "lake" often revealed a conduit that narrowed with depth; sometimes the magma was withdrawn from view[11]. The conduit diameter was estimated to be 5–10 m based on modeling of acoustic signals from bursting gas slugs at the end of each lava fountain episode and drone images captured during repose[10].

In light of these observations as well as our measurements suggesting that the gas erupted during fountaining equilibrated at near-surface pressures, we propose that fountaining was initiated by gas accumulated in a shallow, magma-filled cavity during the repose period. Collapse of a foam layer into the conduit could be triggered by dynamic pressure instabilities induced by slug ascent, as experiments have shown that pressures in the wake of slugs ascending through flared tubes decrease by $10^3$–$10^5$ Pa due to the falling liquid film on the walls of the conduit[39,40]. While slight pressure changes are unlikely to trigger fountaining if the conduit is connected to a deeper cavity (>0.5 km), in shallow cavities this becomes feasible due the pressure sensitivity of $H_2O$ degassing at low pressures, which can lead to accumulation of a $H_2O$-rich foam layer at the roof of a shallow cavity (Fig. 2b). While the size and amount of slugs will be greatest soon after the collapse of this foam layer, the lower density of the magma-gas mixture in the conduit during fountaining will depressurize the magma in the cavity, leading to inflow of fresh magma from depth as well as further $H_2O$ degassing. This decompression-driven magma degassing represents the more extended period of lava fountaining following the attainment of peak fountain heights.

Our conceptual model uniting the physical mechanism driving the episodic lava fountaining with the measured gas chemistry at Fagradalsfjall is sketched in Fig. 3. At the start of the repose period (Fig. 3b), the magma contained in the lake, conduit and plastered on the crater walls was erupted and partially degassed during a previous fountaining

episode. The relatively high $H_2O/CO_2$ ratio (~30–50) and low $SO_2/HCl$ ratio (5-10) represents Rayleigh distillation of the crystallizing magma, with higher solubility volatile species such as $H_2O$ and Cl remaining in the melt in contrast to $SO_2$ and $CO_2$, which are rapidly lost from the magma at the surface[32,33]. As gas and melt are supplied from depth into the shallow cavity, gas accumulates to form a foam on the roof of the cavity, and some bubbles leak through the conduit, decreasing the average density of the magma filling the conduit and causing the upper surface of the lava to increase to maintain pressure equilibrium with a constant pressure source (Fig. 3c). In response to attainment of a critical foam thickness and the ascent of a large slug through the conduit, pressure at the base of the conduit falls below the pressure in the cavity, destabilizing the foam layer and triggering lava fountaining (Fig. 3d). Gas erupted during fountaining retains the approach to melt-gas equilibrium acquired in the shallow cavity, with a lower $H_2O/CO_2$ ratio (~10–15) and higher $SO_2/HCl$ ratio (~30–40). As the foam raft drains through the conduit, the magma in the cavity is depressurized and degassed, resulting in ascending slugs that drive more sporadic and shorter fountains. The sharp decrease in the $SO_2/HCl$ ratio records when the measured gas is no longer derived from magma degassing in the cavity but rather from already outgassed lava erupted at the surface. Drain-back of outgassed melt from the crater into the conduit and inflow of "fresh" melt from the dyke into the shallow cavity recharge the system and the cycle repeats.

We suggest that foam collapse causes fountains to reach their maximum heights soon after the onset of fountaining, and that fountaining is sustained following the attainment of peak fountain heights because fountaining depressurizes the conduit and cavity and promotes further degassing of melt in the underlying cavity, acting as a positive feedback. As our model assumes that the rapidly ascending gas slugs preserve an approach to gas-melt equilibrium acquired at shallow depth, the decrease in the $H_2O/CO_2$ ratio from ~15 to ~8 (Fig. 3c) and increase in the $SO_2/HCl$ ratio from ~25 to ~45 (Fig. 3d) during fountaining could represent gas slugs sourced from progressively greater depths in the cavity throughout a fountaining event. Alternatively, this compositional change could result from slower-diffusing $SO_2$ and $CO_2$ entering later-formed bubbles[34]. At the end of fountaining, the sporadic nature of the spattering indicates progressive outgassing of the magma, which results in more Strombolian-type eruptive behavior.

Although our data do not allow us to visualize the geometry of the shallow magma plumbing system, in Fig. 3 we draw the shallow cavity located in between vent 1 and vent 5, consistent with the model of Eibl et al. [11]. Due to the proximity of these vents, which are about 100 m apart, mechanical or thermal erosion of a particularly weak lithological unit may have created a shallow cavity. A possible analog for such a feature is the Thríhnúkagígur lava cave (SW Iceland), a ~60 m deep cylindrical conduit of ~8 m diameter that narrows with depth to a lung-shape hollow with dimensions of ~80 m along-strike (of the associated fissure) and ~20 m across-strike[41,42]. If a similar cavity existed beneath Fagradalsfjall during the eruption, then we envisage flow between it and the vent to be regulated by a flared-nozzle, as shown in Fig. 3. Based on an estimated surface gas emission of $4 \times 10^5$ $m^3$ per fountaining event at atmospheric pressure (see Methods), then at 1–2 MPa, where we calculate gas separation to occur in the shallow cavity, the volume of the erupted gas is on the order of 20,000–40,000 $m^3$, comparable with the estimated volume of the cavity at Thríhnúkagígur[41,42]. However, since according to our model the gas derived from foam collapse only drives the initial period of fountaining when maximum fountain heights are attained, the gas contained in the foam layer is only a fraction of the total erupted gas. The remainder of the gas is derived from further decompression of magma in the cavity during fountaining.

Our conceptual model combines elements of both the rise-speed dependent (RSD) and collapsing foam (CF) models for lava fountaining

in the framework of pressure fluctuations acting on a lava lake-conduit-cavity system. Similar to the RSD model, Hawaiian style lava-fountaining results due to the relatively rapid ascent rate of the magma, which allows coupled gas-melt flow to the near surface. Similar to the CF model, the fountaining periodicity reflects the dynamics of the collapse and growth of a foam layer at the roof of a cavity. Due to the shallow depth of the cavity, which permits $H_2O$ degassing and gas-melt separation within, the pressure drop resulting from slug ascent can trigger collapse of this layer and the onset of fountaining. While the fountaining periodicity was remarkably consistent throughout the measurement period on 5 May (Fig. 1), the duration of a cycle (including fountaining and repose) varied between 3 and 20 min through the six weeks of cyclical behavior, with a trend towards lengthening period over time[11]. Assuming the physical mechanism we have described is pertinent to the entire episodic fountaining stage, we speculate that the observed increase in the duration of fountaining/repose cycle reflects an increase in the cavity volume and/or conduit diameter with time.

Our gas chemistry data suggest that the much shorter period fluctuations observed at Fagradalsfjall reflect a shallow cavity underlying the eruptive vent, in contrast to the episodic lava fountains at Etna or Kīlauea, for which much longer repose periods indicate control by deeper processes associated with upper crustal magma reservoirs. This type of eruptive style, intermediate between fountaining and Strombolian, may be more common than previously recognized, as the critical elements to produce these types of episodic fountains appear to be: (i) a lava lake connected to a shallow cavity via a conduit, and (ii) relatively low magma supply rates, which preclude continuous fountaining activity. Accordingly, further research into the geometry of the shallow magma plumbing system in basaltic volcanoes is needed to understand the applicability of this model to other settings. More broadly, our dataset highlights how measuring the composition of the erupted volatile phase on fine temporal and spatial scales can provide important insight into the physical and chemical processes governing eruption dynamics.

## Methods
### OP-FTIR measurements
Open-path FTIR spectra were collected on 5 May 2021 from ~300 m from vent 5 (Supplementary Materials, Fig. S1). The open-path spectra were collected with a MIDAC FTIR spectrometer equipped with a liquid nitrogen-cooled mercury cadmium telluride (MCT) detector and 3-inch Newtonian telescope with a 10 mrad field of view. Interferograms and single beam spectra were collected at 0.5 cm$^{-1}$ resolution approximately every 2 s. Examples of collected spectra are shown in Fig. S2 (Supplementary Materials). We determined the column amount of gases contributing to the measured spectra using the Reference Forward Model[43] to simulate absorptions of target volcanic and atmospheric gas molecules in a specified spectral range using line parameters taken from the HITRAN database[44]. The model replicates a two-layer atmosphere with different temperatures for both atmospheric and volcanic gas species. The volcanic gas temperature (an integrated temperature of the atmosphere and volcanic gases between the IR source and the instrument) was set to 280 K, reflecting rapid cooling of the volcanic gas as it mixed with the atmosphere. The calculations were conducted using the software FTIR FIT[45]. The wavebands selected for obtaining retrievals were 2020–2150 (for $CO_2$, CO, $H_2O$), 2410–2550 ($SO_2$), 2600–3000 (HCl) and 4000–4100 cm$^{-1}$ (HF). Laboratory experiments using similar equipment (the same interferometer model) indicate that retrieval accuracies of better than 5% can be achieved[46]. Gas ratios were obtained for each measurement, and correction for ambient air $H_2O$, $CO_2$, and CO column amounts were performed spectrum-by-spectrum based on retrieved methane abundance, which is used as a proxy for the optical distance (which is shorter during fountaining; see Fig. S3). Using the computed distance,

we corrected CO, $CO_2$ and $H_2O$ column amounts for each measurement assuming an atmospheric CO, $CO_2$ and $H_2O$ abundance of 0.065, 420 ppm, and 4150 ppm, respectively (the latter roughly corresponding to a relative humidity of 50% at a pressure of 985 hPa and temperature of 276 K, as measured from a nearby monitoring station). Note that the calculated gas composition is sensitive to the assumed background concentrations of CO, $CO_2$ and $H_2O$. The raw and corrected column amounts for $H_2O$ and $CO_2$ are shown in Fig. S4. The ratios were used to calculate the abundance of each measured gas (Table 1). Note that the resulting calculated total gas composition does not account for minor species including $H_2$ and $H_2S$ that could not be detected with our equipment. We consider that gas measurements are a good proxy for instantaneous gas composition at the vent, though there must be some uncertainty arising from integration along the optical path. To account for temporal latency, the measured composition of gas emitted during fountaining and repose periods used to calculate the gas molecular compositions shown in Table 1 emphasizes measurements made later in each respective period. The retrieved column amounts for all major gases used to calculate the compositions given in Table 1 are shown in Fig. S5.

### Models of magma degassing
The degassing behavior of the Fagradalsfjall melt was modeled using D-Compress[23] using the system C-O-S-H. D-Compress requires the initial pressure, temperature, $fO_2$, $CO_2$ and $H_2O$ contents of the magma as inputs, as well as the melt composition. The program outputs the initial S content and models each of these parameters ($CO_2$–$H_2O$–S–$fO_2$) during decompression. Derived eruption temperatures based on major element concentrations in tephra glasses produced during the lava fountaining are ~1200 °C, which was used for our modeling. The melt composition was set based on measurements of an airfall tephra erupted on April 21-22, 2021 (ref. 18). We assume $fO_2$ conditions corresponding to ΔNNO = −0.7 based on olivine-spinel pairs[18]. Initial volatile contents of the depleted melt (erupted in March) estimated from the most volatile-rich melt inclusions reported in Halldórsson et al. [18]. indicate initial $CO_2$, $H_2O$ and S contents of 5500, 1500 and 1150 ppm, respectively. For the more enriched melt that erupted during the lava fountaining, trace element proxies show an increase in volatile load. The trace element concentrations and inferred initial volatile contents are shown in Table S1. Trace element proxies were used to estimate the initial volatile loads for $CO_2$ (ref. 47), $H_2O$ (ref. 48), and S (ref. 49). A S/Dy ratio of 350 (similar to Holuhraun[50]) reproduces the undegassed S contents of the most evolved melt inclusions. For the more enriched melt, we assume a similar redox state, and initial $CO_2$, $H_2O$ and S contents of 8000 ppm, 3400, and 1150 ppm. We used the composition of the enriched melt for the magma degassing calculations shown in Fig. 2. Sensitivity analyses (Fig. S6) suggest that the initial water contents of the depleted melt are too low to account for the measured $H_2O/SO_2$ ratios but could be consistent with the measured $H_2O/CO_2$ and $CO_2/SO_2$ ratios assuming a lower equilibration pressure (0.1–0.6 MPa).

### Gas flux estimates
Ultraviolet spectroscopy and windspeed measurements indicate an $SO_2$ flux of 51 ± 19 kg s$^{-1}$ throughout the pulsatory lava fountaining in late April to early May but did not resolve fountain and repose emissions. This measured gas flux fits very well with the value calculated based on the time-averaged lava discharge rate (TADR) and melt inclusions. Melt inclusions indicate initial complements of around 1150 ppm S (2500 ppm $SO_2$) by mass (Supplementary Materials), while matrix glasses contain around 250 ppm $SO_2$ by mass[18]. For the observed lava effusion rate of ~3 × 10$^4$ kg s$^{-1}$ (based on a volumetric TADR of 11.4 ± 0.5 m$^3$ s$^{-1}$; ref. 51), we would expect an $SO_2$ flux of 57 kg s$^{-1}$, indistinguishable from the measured value considering uncertainties in atmospheric scattering, wind speed,

differential digital elevation model (DDEM) processing, and magma vesicularity. Since (i) most of the gas emission occurs during fountaining (Fig. 1), (ii) the repose and fountain durations are approximately equal (Fig. 1), (iii) the measured fountain gas contains ~6 wt% $SO_2$ (2 mol% converted to weight proportion; Table 1), and (iv) the fountain duration is of ~4 min, we estimate the total volume of gas released per cycle is ~$4 \times 10^5$ m$^3$ (for a gas density of 1.3 kg m$^{-3}$ at atmospheric pressure).

## Acoustic measurements

The duration of fountaining and repose periods was derived from continuous acoustic measurements. The acoustic data are from processing of a four-element array installed approximately 800 m NW of vent 5 for the purpose of monitoring the eruption[10]. The 70 m aperture array was equipped with InfraBSU V2 infrasound sensors (flat response from 0.1 to >40 Hz; ref. 52) connected to DiGOS DATA-CUBE$^3$ digitisers recording data at 200 Hz. To identify coherent signals from eruptive activity in vent 5, the data was processed with a least-squares beamforming algorithm[53] using 10 s moving windows with 50% overlap. Signals associated with eruptive activity were identified in each time window using back-azimuths from 122 to 162°, trace velocities of 250–400 m s$^{-1}$, and a median cross-correlation maximum (MdCCM) greater than 0.5. For this array, we estimated a theoretical uncertainty in back-azimuth and trace velocity estimates of 4–5° and 18–22 m s$^{-1}$, respectively.

## Video recordings

Fountain heights were determined from video recordings. Movie S1 shows a time-lapse (64x) video of RÚV (Ríkisútvarpið, Iceland's national public service broadcasting organization) camera footage during the FTIR monitoring period on 5 May 2021. Fountain heights shown in Fig. 1a were calculated based on this video following the method described in ref. 10. Briefly, the height of the fountain was inferred from each frame by transforming it from a RGB to a LUV colourspace[54]. As the U component has high values for the redbrown tones visible in the lava fountain during the day, the pixel with the highest U value in each row of the frame was selected, giving a vertical profile of the fountain based on color (bottom panel in Movie S1). The height in the image is then converted to a height in geographic space by projecting the coordinate (vent Easting, vent Northing, fountain height above sea level) into image coordinates and minimising the difference in the projected row number with that retrieved from the color information in the frame. The measurement therefore rests on the assumption that the fountain top is directly above the vent.

## Reporting summary

Further information on research design is available in the Nature Portfolio Reporting Summary linked to this article.

# Data availability

Fitted and corrected column amounts for each gas species obtained from the measured spectra are provided in the Source Data file. This file additionally contains the full time series of calculated gas composition, range (distance to infrared source), calculated fountain height, and signals consistent with lava fountaining derived from the acoustic measurements. Source data are provided with this paper.

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

## Acknowledgements

We thank the Icelandic Ministry of Environment, Energy and Climate (Umhverfis-, orku- og loftslagsráðuneytið; URN) for funding the volcano monitoring efforts at Fagradalsfjall, and Prof. Evgenia Ilyinskaya (University of Leeds) for acquiring the FTIR spectrometer used in this study. We thank Sveinbjörn Steinþórsson for providing help with the eruption monitoring efforts at the University of Iceland and the Department of Civil Protection (Almannavarnir) for providing help and access to the eruption site. S.W.S. thanks Prof. Eva Eibl (University of Potsdam) for stimulating discussions and insight into the physical mechanisms driving intermittent lava fountaining at Fagradalsfjall.

## Author contributions

S.W.S, M.A.P., C.O., R.K. and A.S. collected, processed, and analysed the OP-FTIR data. S.W.S performed the modeling of magma degassing, with help in model set-up and interpretation from E.B., C.O. and A.S. O.D.L. collected, processed, and analyzed the acoustic measurements. T.B. collected, processed, and analysed the data for lava fountain heights. S.W.S, M.A.P., C.O., A.W.W. and O.D.L. developed the interpretation for the physical mechanism driving intermittent fountaining. All co-authors contributed to the manuscript.

## Competing interests

The authors declare no competing interests.
