## [Peer Review File · Nature Communications]

Near-surface magma flow instability drives cyclic lava fountaining at Fagradalsfjall, IcelandREVIEWER COMMENTS

Reviewer #1 (Remarks to the Author):

Review of “Near-surface magma flow instability drives cyclic lava fountaining at Fagradalsfjall, Iceland” for Nature Communications

Review by Ed Llewellyn.

Summary:

This manuscript presents high time-resolution gas geochemistry data for a period of cyclic lava fountaining at the Fagradalsfjall eruption in May 2021. The data presented are spectacular, and data analysis is careful and thorough. The data provide a unique insight into the processes that drove this unusual fountaining activity, and the manuscript presents some interpretations of those processes in terms of subsurface degassing, outgassing, and fluid dynamic behaviour. The manuscript is likely to be of significant interest to the volcanology community and, given the intense interest in this recent, high-profile eruption, to researchers and others beyond that disciplinary community. This will be helped by the fact that the manuscript is well written, and generally presents the story in an engaging and accessible fashion. I recommend publication, if the authors can address a couple of issues, outlined below.

Major comments:

1. *The physical model.*

The manuscript presents a physical model for the origin of cyclic fountaining behaviour, starting on line 114, and summarized in Figure 3. There are aspects of the model that are not adequately developed, and that are not physically justified. In particular, the treatment of the physics of the pressure and flow balances in the shallow conduit is inadequate, and there are parts of it that do not make sense to me.

The model hinges on the idea of counterflow in the conduit, inspired by (or at least leaning on) the conceptual model of Witham et al (2006). However, the explanation for the origin of counterflow in the manuscript relies on a mis-reading or mis-understanding of the Witham model. It is correct (line 125) that an increase in pressure from the lake leads to counterflow, but this is a sudden process that leads to an immediate shift in behaviour from wholesale upward flow to wholesale downward flow. The central idea in Witham’s model is that the introduction of gas bubbles into the conduit, which is in magmastatic balance with the underlying reservoir, decreases the average density of the magma column in the conduit, leading to upward flow. This progressively fills the ‘lake’ (Witham) or flared vent/conduit system (Fagradalsfjall) until magmastatic balance is re-established (or nearly re-established). The progress towards this balance occurs because the gas bubbles have a slip velocity through the magma, such that the lava in the lake (vent) becomes outgassed and therefore the average density of the column increases. The instability that drives counterflow arises because of the difference in cross-sectional area of the lake (flared vent) and underlying conduit. Consequently, a small perturbation near equilibrium, driving some flow downwards from lake into conduit, leads to a small drop in lake level, but a big decrease in the buoyancy of the underlying magma column. This instability is explored in much more detail in Witham and Llewellyn (2006).

The Witham model is not consistent with – or, at least, does not explain the origin of – the proposed progressive counterflow that sets up at the start of fountaining. The authors also appear to imply (line 135) that the counterflow decreases pressure at the base of the conduit. Counterflow of the sort proposed in the Witham model leads to an **increase** in the pressure at the base of the conduit. It is this pressure imbalance (pressure at the conduit base exceeding pressure in the shallow reservoir) that drives the downward flow.

Stepping through the stages identified in the caption to figure 3 (authors' descriptions in red):

Step B: **The level of the lava lake decreases as pressure at the base of the conduit equilibrates with the pressure in the underlying cavity.** This is reasonable and implies that the authors are assuming pressure balance at the interface between conduit and underlying cavity.

Step C: **Ascent of gas bubbles** (from where, the cavity or deeper?) **increases lake level.** This is reasonable – decrease in average density of the column of magma in the conduit would mean that the same pressure in the cavity could support a taller column, filling the lake/vent.

Step D: **The filling of the lake causes gas to accumulate in the conduit and reservoir. Melt counterflow occurs once the vertical pressure gradient is low enough to cause the melt to flow downwards.** I don't see the causal relationship here. The lower average density of the material in the column (and maybe in lake, but presumably separated bubble rise would rapidly release gas in the lake) would decrease the pressure gradient (i.e. rate of increase of pressure with depth) but I don't see any reasonable way that this would cause downward flow of the magma. The decreased pressure gradient would mean that the pressure at the base of the conduit is lower than it would be if there were no gas bubbles in the magma column to decrease its density. This means that the base of the conduit is no longer in pressure equilibrium with the underlying cavity, so magma (and gas) would tend to rise upwards from the cavity, into the conduit. This also means that it is hard to see how **'filling of the lake causes gas to accumulate in the ... reservoir'**. In fact, filling of the lake resulting from ascent of gas bubbles (step C) would be associated with upward flow from the reservoir, not gas accumulation within it. In detail, it matters if the pressure in the cavity is assumed to be constant – perhaps buffered by lithostatic pressure at the cavity walls, or by pressure balance with a deeper storage – or if the authors imagine that the cavity has a volume that is effectively fixed over the timescale of fountain onset. Regardless, I don't see any mechanism by which a decreased pressure gradient in the conduit would cause melt counterflow.

Step E: **Counterflow causes accumulated gas to empty from the reservoir.** This is reasonable but requires a convincing mechanism for driving counterflow. **Lower pressure at the base of the conduit promotes gas exsolution via magma decompression.** Agreed, but the mechanism for lowering the pressure is not explained.

Step F: **The progressive outgassing of the magma in the reservoir causes a gradual waning in the intensity of the fountaining, until dense, outgassed melt drains back into the conduit and the cycle repeats.** I agree with the bulk of this, but there is no mechanism here to explain why the cycle repeats.

So, for me there are two issues that are not properly addressed:

- a) How counterflow is setup and how this relates to the increase in the height of the lava lake. Related to this, the Witham model is not represented accurately and does not provide a mechanism for the onset of counterflow – at least not in the way described in the text.
- b) How the cycle is reset.

2. Interpretations of the gas geochemistry

I find some inconsistencies in the discussion around Figure 2, which may result from my lack of expertise in gas geochemistry. Nevertheless, if I miss the point here, others may too. It seems that it is first argued (paragraph starting line 76) that the H₂O/CO₂ ratios are consistent with open system

equilibration at 20-100 MPa, then closed system to the surface. This evidence is used to argue for substantial loss of CO₂ at a depth of ~4 km. However, the next paragraph argues that the ratios favour open-system equilibration at ~2 MPa, rather than closed-system from 20 or 100 MPa to the surface. This latter argument does indeed seem to be supported by the ratios shown in Figure 2B-E. My issue is that I therefore don't see the rationale for equilibration at 20-100 MPa, if the closed-system path from 20 or 100 MPa is rejected on the basis of the ratios in Figure 2B-E. I do see that there is a problem in explaining where the excess CO₂ goes. If the system degasses in open system to 2 MPa, but then releases gases in equilibrium ratios at that depth, where is the excess that was released on the way to 2 MPa? A major outgassing event at 20-100 MPa, in which CO₂ is (how?) sequestered into a geothermal system gets rid of some of the problem, but what happens to the gas released between 20 (or even 100) MPa and the surface? I don't expect the authors to have all the answers, but this is a lot of gas to explain away, and currently there is no attempt to do so. If the rationale behind the deeper equilibration is actually to resolve (at least some of) the excess CO₂ problem, then that should be stated much more explicitly. At the moment the release of CO₂ comes first (argued from the closed-system ratios) and the geothermal sequestration is presented as consistent with this scenario. Something doesn't quite make sense here – at least to me.

A minor related point, which again may stem from my lack of expertise in gas geochemistry: why are the 20 MPa open system and 100 MPa open system curves different? If a melt equilibrates open system at 100 MPa, then stays open system to the surface, why would it be producing a gas with different composition at 20 MPa than a melt that had followed a closed path to 20 MPa, then equilibrated at 20 MPa? The difference is small, but I don't understand it's origin. I just want to check that there is a good reason for this, and it is not an indication of an error, or of some additional processing step that is not properly explained.

Minor comments:

Line 5: 'widespread' is a bit of an odd word choice and might confuse non-idiomatic English speakers into thinking this means that lava fountains spread wide. I suggest it's easier just to use 'common'.

Line 41: Slight grammar issue. Either change to "...beg the question: what...?" or drop the question mark at the end of the sentence. I prefer the former.

Line 49: Focus is on H₂O and CO₂, but do the other species support the same interpretations? Is there anything that can be learnt about residence times in the shallow reservoir from the changes in abundances of the slower diffusing species?

Line 56: The data cover 16 cycles, which I think is plenty to characterize this particular portion of the eruption, but is there any evidence to either support or reject the mechanism proposed more generally over the eruption?

Figure 1: These are really lovely data. However, the compressed axes and non-vector nature of the plots mean that it's hard to see the detail. In plots C-H, presumably the y-axes that include 0 are linear scale, but E is ambiguous, and could be logarithmic (like B). It would be good to include a higher resolution, vector graphics version of this plot in the SI. Ideally, this would be accompanied by all of the raw data (e.g. as a csv file) so that readers can replot and investigate this wonderful dataset themselves.

Line 78: How sensitive are findings to the assumed values of initial volatile contents? Does varying within uncertainties change the picture in a meaningful way?

Line 103: The high H₂O/CO₂ ratios during repose are explained as resulting from disequilibrium processes, based on the difference between measured residual volatile concentrations in eruption products, and calculated equilibrium values. This makes sense and, in particular, the relative deficit in CO₂ in the gas, and excess in the products, can be explained in terms of relatively low diffusivity of CO₂ compared with H₂O. To what extent could disequilibrium degassing explain the gas trends observed during fountaining? Could it affect the depth or degree of any re-equilibration inferred? Presumably disequilibrium degassing would remove some of the excess CO₂ problem, particularly if coupled with convection processes that could sequester magma back down before CO₂ degassing has gone to completion (i.e. H₂O could have time to degas and separate, but CO₂ not).

132: Is the lava plastered on the walls sufficient to produce the amount of gas required to achieve the observed flux at the required ratios?

133: The decrease in H₂O/CO₂ 'as the lake fills' is inferred to occur before fountaining starts. How is the CO₂ rich gas released? Presumably this must be from partly stagnated magma that infilled conduit between shallow storage and vent. Are the ratios and fluxes observed consistent with this? If not, what is happening? Is the idea that the excess is actually vanguard gas from the shallow storage? This could matter for the physical interpretation.

176: 'shallow cavity at depth' is an awkward construction.

229: It's not clear to me which values of initial volatile concentrations were used in the calculations. The text gives values for 'primitive melt' and different values for 'enriched melt' but doesn't unambiguously state which were used for the D-Compress and SolEx calculations.

References:

Witham, F. and Llewelin, E.W., 2006. Stability of lava lakes. *Journal of Volcanology and Geothermal Research*, 158(3-4), pp.321-332.

Reviewer #2 (Remarks to the Author):

Review of near-surface magma flow instability.....

Summary

This is an excellent addition to the literature of lava fountaining eruption. The gas data was gathered at a temporal resolution equal or better than any previous spectroscopic study. The authors document for the first time (globally) a clear two-stage pattern of magma outgassing. The one area that needs to be enhanced to make the paper suitable for Nature is a much more powerful discussion/conclusion section. The data presented here must be used to evaluate critically the competing conceptual models for lava fountaining described in the early part of the text. To my mind the data presented here makes it clearer that one of two proposed models for lava fountaining, annular flow, is totally untenable. The authors should strengthen the paper by stating unequivocally, their position with respect to the merits of the two competing models for Fagradalsfjall.

General comments

1) I am confused reading the text from line 130 onwards and comparing it to the caption for Figure 3. Where is the 'shallow cavity' mentioned in the text visible on the cartoons on the left of Figure 3. The caption for cartoon B talks about the 'cavity' underlying the base of the conduit. Does this mean the cavity is the wide area at the bottom of each sketch and if so is it part of the reservoir? Or is the reservoir significantly deeper and not shown on the cartoons? It would help a lot if an approximate depth could be stated for this cavity.

2) In the cartoons labelled B and C. (Figure 3) there seems to be a lava flow perched above the cone on the right hand side, which is well above the level of lava in the vent and so the flow appears to be perched and decoupled from source. Is this intended?

What are the noteworthy results?

The key result is the recognition of changing gas chemistry with time during fountaining episodes, which reflects a progressive decrease in the depth at which magma outgassing is occurring.

Will the work be of significance to the field and related fields?

Yes, if the authors strengthen up the conclusion section. This should force models to critically examine all pre-existing models for the dynamics of magma ascent accompanying lava fountaining. In addition, there have been few groups prepared to make volatile measurements of lava fountaining on fine temporal and spatial scales and this study should encourage others, particularly in the USA and Italy, to make similar measurements.

Does the work support the conclusions and claims, or is additional evidence needed?

It is the reverse, there needs to be more forceful conclusions, based on these excellent data.

Are there any flaws in the data analysis, interpretation and conclusions? No.

Is the methodology sound? Does the work meet the expected standards in your field? Yes.

Is there enough detail provided in the methods for the work to be reproduced? Yes.

The manuscript is clearly written and well constructed. Figure 1 is excellent and is the heart of the paper. Figure 3 is very crowded and confusing particularly with respect to the 'cavity' and the 'reservoir'. It could do with some revision to increase the size of the key cross-sections.

Minor Comments

These are made on the attached copy of the manuscript.

Reviewer #3 (Remarks to the Author):

This is an interesting paper centered on an impressively cyclic gas chemistry dataset from the

Fagradalsfjall eruption of 2021.

The authors collected FTIR spectra during repeating lava fountaining events and found repeating patterns of gas chemistry and eruptive behavior. The authors then link the observational data to models of degassing behavior to elicit a model of the behavior driving the cyclic fountaining.

This is a nice dataset and I believe the paper will be a worthy contribution to Nature Communications following some edits and revisions.

My main concern currently is that while the authors outline the observed gas chemistry cycles and how they relate to modeled gas behavior and composition, they then describe three existing models for lava fountaining and identify one that fits with their observations, but fail to adequately discount the other two that they've already introduced. They don't include enough information about how they've drawn on their data to arrive at their conceptual model. In particular, given some similarities to their conceptual cartoon (Figure 3) and the 'foam collapse' model that is introduced but not mentioned again, I'd imagine that readers would want to know why the authors feel the 'pressure cycles' model is a better fit to the chemistry data than is the 'foam collapse' model. The discussion should explain how the other models would result in chemical results incongruous with their observations. As currently written, even if the conclusions about the best fitting model are correct, it feels as if the discussions and explanations of how those conclusions were reached are lacking.

I also feel that the manuscript would benefit from the removal of the use of the SolEx model. SolEx is only applicable to systems with an oxygen fugacity of greater than +0.5 units above the NNO buffer; this system is more than a log unit below that limit. The authors themselves mention doubts about SolEx given that it does not allow for inclusions of fractional degassing in the modeling. Ultimately chlorine behavior doesn't appear to bear on results and conclusions much; the authors (rightly) focus on the behavior of ratios involving H₂O, CO₂, and SO₂. Given its minor role in the paper and questions regarding the appropriateness of the model, perhaps SolEx should be omitted. There should be a discussion of/justification for still using SolEx, if it is to be kept in the paper.

The other main thing that would improve the clarity of the paper is expanded discussion of what is happening during the 'repose' periods. Particularly around lines 43-59, the text mentions 'repose' periods, but what is happening in the vent during that time? I don't think it's actually described here. Is there lesser fountaining out of sight (roiling)? A stagnant lava lake? A drained conduit passively releasing gas? Without that description, it's not clear what the 'repose' gas represents. Similarly, lines 58-59 mentions lava being the IR source during fountaining, but what is the IR source during repose? Overall, information regarding the repose measurements is lacking and needs to be included in a revised version.

Other, smaller suggestions and issues are brought up for specific lines/items below.

I thank the authors and the editors for the invitation to review this paper.

- Lines 27/30 – Is there a reason for use of cyclic vs cyclical? If they are meant to have the same meaning, being consistent with one seems preferred.
- Line 41 – This should mention comparable melts, not magmas (magma encompasses the gas phase along with crystals and melt; in this case, the bulk magma is indeed different given that the shallow-sourced eruptions have already lost CO₂).
- Lines 64-69 – I believe readers would benefit from being able to see the mentioned gas ratios in table form, either in addition to or instead of the in-text comparisons of differences in values between repose and fountaining (and in addition to the running time series plots in Figure 1). Perhaps this could be added to Table 1.
- Figure 1 – Intensity decreases during the repose intervals (<10k, and even <5k in some cases). I understand that units are arbitrary, and may vary from instrument to instrument, but these values seem quite low (perhaps problematically so?). How does low intensity affect noise in your spectra and thus in the retrievals? Particularly of trace species like HF and CO? Can the authors quantify error or confidence levels for the retrievals, and does it differ between repose and

fountaining? This would help confirm for readers that changes in gas chemistry are real rather than artifacts of possible noise in spectra with low intensity.

- Lines 80-83 – Can the authors add approximate equivalent depths for the pressures here, as they do in line 85?
- Line 83 – Here the paper mentions the ratio as being by volume. Plots in Figures 1-3 are molar ratios. Yes, they work out to the same for gases, but best to be consistent with language and change in-text mentions to molar to match the figures.
- Line 93 – This has the figure letters as lowercase, but in the figures and captions, they are capital letters. Should they be the same? I'm not positive about journal style requirements.
- Lines 107-109 – The H₂O and CO₂ being higher than predicted can make sense based on finite diffusion times, but it's curious that there's less S in the glass than predicted given that it's a slow diffuser. Should this be explored more? Lerner et al 2021 showed S in Kīlauea 2018 matrix glasses more along the lines of your expected 700 ppm. I wonder about the discrepancy.
- Line 146 – This relates to the comment above about lines 43-59; this mention of a lava lake comes out of nowhere. Addressing the lack of information about the repose periods earlier in the paper will make this sentence/mention less surprising.
- Line 151 – The text cites other studies that arrive at this conduit diameter, but it may be worth explicitly (and briefly) mentioning here how that diameter was determined in those studies.
- Lines 169-171 – This is concluding a progressive decrease in source volatile content over time. It would be helpful if the authors could discuss/hypothesize about why this might be.
- Line 230 – The FTIR data was later than this tephra eruption date. Was there any documented change in erupted composition over the course of the eruption that might call for using a different starting composition?
- Line 257 and following section – How did the acoustic measurements compare to visual assessments of fountaining versus repose? Were the acoustic data necessary (and would they be for future studies) or could that have been done with the video that was being scrutinized for fountain heights already?
- Figure S5 – Even after correction, there seems to be somewhat of an issue with the fountain data for H₂O/SO₂. Possibly also in the H₂O/CO₂ plot. Either a curve at lower concentrations, or an offset from the origin (incomplete correction?) such that the data distribution doesn't fit will amidst the linear slopes. Is this real? A problem with the data? How has that affected the apparent slopes/bulk ratios (as in the plots of this figure)?
- Figure 3a – The purple star doesn't look purple and is impossible to recognize with the skewed perspective unless very zoomed in. I suggest changing the wording and the symbology for the current size/configuration of the figure.

We thank the reviewers for their constructive reviews. Our responses are indicated with red text.

Reviewer #1 (Remarks to the Author):

Review of “Near-surface magma flow instability drives cyclic lava fountaining at Fagradalsfjall, Iceland” for Nature Communications

Review by Ed Llewellyn.

Summary:

This manuscript presents high time-resolution gas geochemistry data for a period of cyclic lava fountaining at the Fagradalsfjall eruption in May 2021. The data presented are spectacular, and data analysis is careful and thorough. The data provide a unique insight into the processes that drove this unusual fountaining activity, and the manuscript presents some interpretations of those processes in terms of subsurface degassing, outgassing, and fluid dynamic behaviour. The manuscript is likely to be of significant interest to the volcanology community and, given the intense interest in this recent, high-profile eruption, to researchers and others beyond that disciplinary community. This will be helped by the fact that the manuscript is well written, and generally presents the story in an engaging and accessible fashion. I recommend publication, if the authors can address a couple of issues, outlined below.

Major comments:

1. The physical model.

The manuscript presents a physical model for the origin of cyclic fountaining behaviour, starting on line 114, and summarized in Figure 3. There are aspects of the model that are not adequately developed, and that are not physically justified. In particular, the treatment of the physics of the pressure and flow balances in the shallow conduit is inadequate, and there are parts of it that do not make sense to me.

The model hinges on the idea of counterflow in the conduit, inspired by (or at least leaning on) the conceptual model of Witham et al (2006). However, the explanation for the origin of counterflow in the manuscript relies on a mis-reading or mis-understanding of the Witham model. It is correct (line 125) that an increase in pressure from the lake leads to counterflow, but this is a sudden process that leads to an immediate shift in behaviour from wholesale upward flow to wholesale downward flow. The central idea in Witham’s model is that the introduction of gas bubbles into the conduit, which is in magmastatic balance with the underlying reservoir, decreases the average density of the magma column in the conduit, leading to upward flow. This progressively fills the ‘lake’ (Witham) or flared vent/conduit system (Fagradalsfjall) until magmastatic balance is re-established (or nearly reestablished).

The progress towards this balance occurs because the gas bubbles have a slip velocity through the magma, such that the lava in the lake (vent) becomes outgassed and therefore the average density of the column increases. The instability that drives counterflow arises because of the difference in cross-sectional area of the lake (flared vent) and underlying conduit. Consequently, a small perturbation near equilibrium, driving some flow downwards from lake into conduit, leads to a small drop in lake level, but a big decrease in the buoyancy of the underlying magma column. This instability is explored in much more detail in Witham and Llewellyn (2006).

The Witham model is not consistent with – or, at least, does not explain the origin of – the proposed progressive counterflow that sets up at the start of fountaining. The authors also appear to imply

(line 135) that the counterflow decreases pressure at the base of the conduit. Counterflow of the sort proposed in the Witham model leads to an **increase** in the pressure at the base of the conduit. It is this pressure imbalance (pressure at the conduit base exceeding pressure in the shallow reservoir) that drives the downward flow.

Stepping through the stages identified in the caption to figure 3 (authors' descriptions in red):

Step B: The level of the lava lake decreases as pressure at the base of the conduit equilibrates with the pressure in the underlying cavity. This is reasonable and implies that the authors are assuming pressure balance at the interface between conduit and underlying cavity.

Step C: Ascent of gas bubbles (from where, the cavity or deeper?) increases lake level. This is reasonable – decrease in average density of the column of magma in the conduit would mean that the same pressure in the cavity could support a taller column, filling the lake/vent.

Step D: The filling of the lake causes gas to accumulate in the conduit and reservoir. Melt counterflow occurs once the vertical pressure gradient is low enough to cause the melt to flow downwards. I don't see the causal relationship here. The lower average density of the material in the column (and maybe in lake, but presumably separated bubble rise would rapidly release gas in the lake) would decrease the pressure gradient (i.e. rate of increase of pressure with depth) but I don't see any reasonable way that this would cause downward flow of the magma. The decreased pressure gradient would mean that the pressure at the base of the conduit is lower than it would be if there were no gas bubbles in the magma column to decrease its density. This means that the base of the conduit is no longer in pressure equilibrium with the underlying cavity, so magma (and gas) would tend to rise upwards from the cavity, into the conduit. This also means that it is hard to see how 'filling of the lake causes gas to accumulate in the ... reservoir'. In fact, filling of the lake resulting from ascent of gas bubbles (step C) would be associated with upward flow from the reservoir, not gas accumulation within it. In detail, it matters if the pressure in the cavity is assumed to be constant – perhaps buffered by lithostatic pressure at the cavity walls, or by pressure balance with a deeper storage – or if the authors imagine that the cavity has a volume that is effectively fixed over the timescale of fountain onset. Regardless, I don't see any mechanism by which a decreased pressure gradient in the conduit would cause melt counterflow.

Step E: Counterflow causes accumulated gas to empty from the reservoir. This is reasonable but requires a convincing mechanism for driving counterflow. Lower pressure at the base of the conduit promotes gas exsolution via magma decompression. Agreed, but the mechanism for lowering the pressure is not explained.

Step F: The progressive outgassing of the magma in the reservoir causes a gradual waning in the intensity of the fountaining, until dense, outgassed melt drains back into the conduit and the cycle repeats. I agree with the bulk of this, but there is no mechanism here to explain why the cycle repeats.

So, for me there are two issues that are not properly addressed:

a) How counterflow is setup and how this relates to the increase in the height of the lava lake. Related to this, the Witham model is not represented accurately and does not provide a mechanism for the onset of counterflow – at least not in the way described in the text.

b) How the cycle is reset.

We thank the reviewer for this constructive feedback. We have extensively revised the conceptual model for the fountaining cycle in light of these comments. In the revised manuscript, we do not

suggest that melt counterflow is the trigger for fountaining. Instead, based on the models of Witham et al. (2006) and Witham and Llewelin (2006) relating magmastatic balances in a lava lake-conduit-cavity system, we propose that the trigger for fountaining is decreasing pressure at the base of the conduit as the lava lake fills and the conduit becomes progressively more gas-rich. Fountaining results once the magmastatic pressure at the base of the conduit is less than the pressure at the top of the cavity, which causes gas that has accumulated in the cavity to empty into the conduit. Counterflow still factors into the physical mechanism via drain-back, which happens during fountaining and after the magma filling the shallow cavity has become progressively outgassed. We clearly state that the cycle is reset by drain-back of outgassed melt into the conduit and gradual recharge of gas and melt from below into the shallow cavity.

We hope that these changes address your valid concerns about the use of the Witham et al. (2006) and Witham and Llewelin (2006) models, the role of melt counterflow, and the mechanism by which the fountaining cycle is reset. The relevant section of the text (lines 128-156) now reads:

A further mechanism for episodic activity has been proposed in the context of two-phase flow instabilities in shallow lava lake systems³⁶⁻³⁷. As ascending gas from a shallow magma-filled cavity drives magma from the conduit into the lake, it gradually fills the lake with magma. The increasing pressure from the lava lake increases the amount and size of gas slugs in the conduit, and once a sufficiently large slug rises into the lake, the conduit refills with magma and the pressure at the top of the cavity increases, leading to a total reversal of the flow and partial drainage of the lake. Afterwards, new magma and gas is supplied from below into the lake and the cycle repeats.

Our high-resolution gas chemistry data suggest that fountaining was driven by cyclical decompression of a shallow magma-filled cavity (Figure 3). At the start of the repose period (Fig. 3b), the H₂O/CO₂ ratio gradually increases while the SO₂/HCl ratio gradually decreases, as the magma filling the lava lake and plastered on the crater walls becomes progressively outgassed. As the lava lake fills to close to its maximum level (Fig. 3c), the H₂O/CO₂ ratio decreases and SO₂/HCl increases, reflecting the CO₂- and SO₂-richer gas flowing through the conduit that equilibrated just beneath the surface. As the conduit and lake become progressively more gas-rich, the average density of the magma filling the lake and conduit decreases, and the magmastatic head exerted on the top of the cavity decreases. Once the pressure at the base of the conduit is less than the pressure in the cavity, gas that has accumulated in the cavity during the repose period is released, triggering lava fountaining (Fig. 3d). Fountaining depressurizes the conduit and cavity and promotes further outgassing of melt in the underlying cavity, acting as positive feedback. As gas is sourced from progressively greater depths in the cavity throughout a fountaining event, and there is more time for slower-diffusing SO₂ and CO₂ to enter gas bubbles, the H₂O/CO₂ ratio decreases and SO₂/HCl increases during fountaining (Fig. 3c-d). Eventually, magma in the cavity becomes progressively outgassed, and the sharp decrease in the SO₂/HCl ratio records when the measured gas is no longer derived from the cavity but rather from already outgassed lava. Drain-back of melt from the crater into the conduit and inflow of melt and gas from the dyke into the shallow cavity recharge the system and the cycle repeats.

Figure 1. Sketch of the lava fountaining cycle at Fagradalsfjall. **A.** Three-dimensional perspective of eruption site, with vent 5 (the locus of eruption activity of the time, yellow star) fed by a melt conduit connected to a dyke at ~1 km depth. The cross-sections in **B-E** focus in on an area where a lava lake is connected to magma upflow from depth that supplies melt (red) and gas (blue) through a conduit. **B.** The level of the lava lake decreases as pressure at the base of the conduit equilibrates with the pressure in the underlying cavity. **C.** Ascending gas bubbles from the cavity transport gas and melt to the surface, increasing the level of the lava lake. Increasing pressure from the deeper lake causes more gas to accumulate in the conduit. **D.** Once magmastatic head at the top of the conduit is less than the pressure in the cavity, accumulated gas empties from the cavity into the conduit, leading to slug coalescence and intense fountaining, which feeds surface lava flows. As gas is evacuated from the conduit, melt refills the conduit via drain-back. **E.** Lower pressure at the base of the conduit promotes gas exsolution via magma decompression. The progressive outgassing of the magma in the reservoir causes a gradual waning in the intensity of the fountaining, until dense, outgassed melt drains back into the conduit. The cycle repeats as melt and gas are supplied from depth. The lower panel on the right shows the spectroscopic measurements for one of these fountaining cycles (see Fig. 1), relating **F.** H₂O/CO₂ and **G.** SO₂/HCl to the fountaining cycle.

2. Interpretations of the gas geochemistry

I find some inconsistencies in the discussion around Figure 2, which may result from my lack of expertise in gas geochemistry. Nevertheless, if I miss the point here, others may too. It seems that it is first argued (paragraph starting line 76) that the H₂O/CO₂ ratios are consistent with open system equilibration at 20-100 MPa, then closed system to the surface. This evidence is used to argue for substantial loss of CO₂ at a depth of ~4 km. However, the next paragraph argues that the ratios favour open-system equilibration at ~2 MPa, rather than closed-system from 20 or 100 MPa to the surface. This latter argument does indeed seem to be supported by the ratios shown in Figure 2B-E.

My issue is that I therefore don't see the rationale for equilibration at 20-100 MPa, if the closed system path from 20 or 100 MPa is rejected on the basis of the ratios in Figure 2B-E. I do see that there is a problem in explaining where the excess CO₂ goes. If the system degasses in open system to 2 MPa, but then releases gases in equilibrium ratios at that depth, where is the excess that was released on the way to 2 MPa? A major outgassing event at 20-100 MPa, in which CO₂ is (how?) sequestered into a geothermal system gets rid of some of the problem, but what happens to the gas released between 20 (or even 100) MPa and the surface? I don't expect the authors to have all the answers, but this is a lot of gas to explain away, and currently there is no attempt to do so. If the rationale behind the deeper equilibration is actually to resolve (at least some of) the excess CO₂ problem, then that should be stated much more explicitly. At the moment the release of CO₂ comes first (argued from the closed-system ratios) and the geothermal sequestration is presented as consistent with this scenario. Something doesn't quite make sense here – at least to me.

We are grateful for the chance to clarify this potential point of confusion raised by the reviewer. As noted by the reviewer, a major feature of the presented models is a first stage of loss of deeply exsolved gas at pressures 20-100 MPa, corresponding to depths of ~1-5 km. As shown in Figure 2, if we consider a stage of gas loss at 20 MPa, measured ratios are consistent with closed- or open-system degassing and equilibration at 1-2 MPa. We consider open-system degassing to be likely due to the much more rapid ascent speed of bubbles compared to melt. As evidenced by the roiling lava lake during the repose stages, gas bubbles are ascending through the magma even when the magma isn't erupting. This strongly suggests at least a partial role for open-system degassing in the shallow cavity. We have modified the text in this section in the hope of improving the clarity of the assumptions underlying the degassing model (lines 90-104):

Assuming closed-system degassing during magma ascent followed by a first stage of gas loss at 20–100 MPa, we can reset the dissolved volatile contents of the melt accordingly then model a second stage of closed- and open-system degassing of the now partially-degassed melt to the surface (Fig. 2b-e). This yields much closer correspondence between the modeled and measured gas compositions.

Figures 2b-e show that the measured H₂O/CO₂, H₂O/SO₂ and CO₂/SO₂ ratios during fountaining (grey shading in Fig. 2) are all consistent with low pressure equilibration. In particular, the measured H₂O/CO₂ of 12±3 during fountaining suggests equilibration at a pressure of ~2 MPa for an open-system degassing scenario. While the measured ratios are also consistent with closed-system degassing following the first stage of gas loss, open-system degassing is favored by the more rapid ascent speed of bubbles compared to melt, which results in continual gas separation and loss at such low pressures^{28,30,31}. Thus, we envision a scenario involving continual gas loss through the surface, with the exception of some gas exsolved at ~2 MPa, which accumulates at the top of a shallow cavity during repose intervals and drives the fountaining (see below).

A minor related point, which again may stem from my lack of expertise in gas geochemistry: why are the 20 MPa open system and 100 MPa open system curves different? If a melt equilibrates open system at 100 MPa, then stays open system to the surface, why would it be producing a gas with different composition at 20 MPa than a melt that had followed a closed path to 20 MPa, then equilibrated at 20 MPa? The difference is small, but I don't understand its origin. I just want to check that there is a good reason for this, and it is not an indication of an error, or of some additional processing step that is not properly explained.

We thank the reviewer for pointing out this additional point of confusion. The difference between the two curves for open-system degassing probably arise from slight differences in the redox state of

the magma resulting from differences in the initial volatile content of the melt depending on whether deeply-exsolved gas was lost at 100 MPa or 20 MPa. According to D-Compress, melt that has lost gas at 100 MPa is slightly more reduced than the melt that has lost gas at 20 MPa (DNNO = -0.787 compared to DNNO = -0.783). Due to the low solubility of CO₂ at low pressures, such slight variations in redox could affect the relative proportions of H₂O and CO₂ released during a depressurization step. Indeed, after checking the model calculations, we noted that the solubility of CO₂ within the pressure range of 1-5 MPa calculated by the open-system degassing pathway from 100 MPa was slightly higher than that calculated by the open-system degassing pathway at 20 MPa. This discrepancy accounted for the slightly higher H₂O / CO₂ ratios at a given pressure calculated by the open-system degassing pathway from 100 MPa compared to the open-system degassing pathway at 20 MPa.

Since we believe this to be mainly a numerical artifact rather than a significant difference between the two melts, and is likely to lead to confusion, in the revised manuscript, we have chosen to only retain a single curve in Figure 2 for the open-system degassing calculations (the curve produced by open-system degassing from 100 MPa to the surface).

Minor comments:

Line 5: 'widespread' is a bit of an odd word choice and might confuse non-idiomatic English speakers into thinking this means that lava fountains spread wide. I suggest it's easier just to use 'common'.

The suggestion of the reviewer has been adapted in the revised manuscript.

Line 41: Slight grammar issue. Either change to "...beg the question: what...?" or drop the question mark at the end of the sentence. I prefer the former.

The suggestion of the reviewer has been adapted in the revised manuscript.

Line 49: Focus is on H₂O and CO₂, but do the other species support the same interpretations? Is there anything that can be learnt about residence times in the shallow reservoir from the changes in abundances of the slower diffusing species?

We consider modeling of disequilibrium (diffusion-limited) degassing to be outside of the scope of the manuscript. In terms of residence times, we believe that the duration of the fountaining/repose cycles provides the best constraints on magma residence times in the shallow cavity. However, in the revised manuscript, we added discussion of the possible role of diffusion in producing the changes observed during fountaining on lines 115-118.

While diffusion-limited CO₂ degassing could also explain the elevated H₂O/CO₂ ratios observed during fountaining, the nucleation and storage of H₂O-rich gas bubbles in a shallow cavity would contribute to a closer approach to equilibrium³³.

Moreover, we describe a role for diffusion in explaining the transient changes in gas composition during fountaining on lines 148-151:

As gas is sourced from progressively greater depths in the cavity throughout a fountaining event, and there is more time for slower-diffusing SO₂ and CO₂ to enter gas bubbles, H₂O/CO₂ molecular ratio decreases and SO₂/HCl molecular ratio increases during fountaining (Fig. 3c-d).

Line 56: The data cover 16 cycles, which I think is plenty to characterize this particular portion of the eruption, but is there any evidence to either support or reject the mechanism proposed more generally over the eruption?

In the revised manuscript, we highlight the generality of our proposed conceptual model, and discuss its applicability throughout the episodic fountaining stage of the eruption. The relevant text is on lines 179-198:

Our proposed conceptual model combines elements of both the rise speed dependent (RSD) and collapsing foam (CF) models for lava fountaining in the framework of pressure fluctuations acting on a lava lake-conduit-cavity system. Similar to the CF model, the build-up and release of gas accumulated at the top of a cavity acts as a trigger for the onset of fountaining. However, sustained fountaining does not reflect the near-instantaneous collapse of a foam layer and the establishment of annular flow conditions in the conduit, but rather progressive nucleation and coalescence of bubbles as magma depressurizes and ascends, as proposed by the RSD model. As the duration of a fountaining event depends on the balance between the supply of melt and gas through the shallow cavity and the rate of magma outgassing and drain-back into the conduit, fountaining is episodic rather than continuous because the rate at which melt and gas recharge the cavity is slower than the rate at which the cavity evacuates during fountaining. While the fountaining periodicity was remarkably consistent throughout the measurement period on 5 May (Fig. 1), the duration of a cycle (including fountaining and repose) varied between 3 and 20 min throughout the six weeks of this cyclical behavior, with a trend towards lengthening period over time¹². Assuming the physical mechanism described by our conceptual model is applicable to the entire episodic fountaining stage, an increase in the repose period would suggest a progressive decrease in the rate of deep recharge of melt and gas, as more time would be required to generate the pressure fluctuations necessary to induce fountaining. However, other factors likely come into play, including changes in the conduit radius due to mechanical or thermal erosion¹².

12. Eibl, E. P. S. *et al.* Evolving shallow conduit revealed by tremor and vent activity observations during episodic lava fountaining of the 2021 Geldingadalir eruption, Iceland. *Bull. Volcanol.* **85**, (2023).

Figure 1: These are really lovely data. However, the compressed axes and non-vector nature of the plots mean that it's hard to see the detail. In plots C-H, presumably the y-axes that include 0 are linear scale, but E is ambiguous, and could be logarithmic (like B). It would be good to include a higher resolution, vector graphics version of this plot in the SI. Ideally, this would be accompanied by all of the raw data (e.g. as a csv file) so that readers can replot and investigate this wonderful dataset themselves.

To clarify, we have added the following text to the figure caption "Note that all data in C-H are plotted using a linear scale." In addition, we will provide the data (calculated retrievals as column amounts) as a csv file.

Line 78: How sensitive are findings to the assumed values of initial volatile contents? Does varying within uncertainties change the picture in a meaningful way?

To investigate the sensitivity of model calculations to assumed values of initial volatile contents, we performed additional calculations comparing the primitive (initial CO₂, H₂O and S contents of 5500, 1500 and 1150 ppm, respectively) and enriched (initial CO₂, H₂O and S contents of 8000 ppm, 3400,

and 1150 ppm) melts, and added an additional figure to the supplementary materials showing the results of these calculations.

Figure S6. Sensitivity analysis for degassing calculations of the Fagradalsfjall melt, comparing “primitive” melt (dashed lines) and “enriched” melt (solid lines). Initial volatile compositions for “primitive” and “enriched” melts given in text – the “primitive” melt is distinguished by a lower H₂O content. A. Dissolved CO₂, H₂O, and S concentrations in the melt as a function of pressure. B. H₂O/CO₂, and C. H₂O/SO₂, and D. CO₂/SO₂ after a stage of gas loss at 100 MPa (see text). Closed- and open-system degassing after gas loss at 100 MPa shown with blue and green lines, respectively.

We have added discussion of this figure to the methods section on lines 264-267:

Sensitivity analyses (Figure S6) suggest that the initial water contents of the primitive melt are too low to account for the measured H₂O/SO₂ ratios, but could be consistent with the measured H₂O/CO₂ and CO₂/SO₂ ratios assuming a lower equilibration pressure (0.5-0.6 MPa).

Line 103: The high H₂O/CO₂ ratios during repose are explained as resulting from disequilibrium processes, based on the difference between measured residual volatile concentrations in eruption products, and calculated equilibrium values. This makes sense and, in particular, the relative deficit in CO₂ in the gas, and excess in the products, can be explained in terms of relatively low diffusivity of CO₂ compared with H₂O. To what extent could disequilibrium degassing explain the gas trends observed during fountaining? Could it affect the depth or degree of any re-equilibration inferred? Presumably disequilibrium degassing would remove some of the excess CO₂ problem, particularly if coupled with convection processes that could sequester magma back down before CO₂ degassing has gone to completion (i.e. H₂O could have time to degas and separate, but CO₂ not).

We thank the reviewer for this comment. The potential for disequilibrium degassing throughout all stages of magma degassing (during magma ascent, degassing in the shallow cavity, and at the surface) is source of uncertainty in our modeling calculations, which assume equilibrium at all pressures. Pichavant et al. (2013) show that CO₂ degassing is diffusion-limited in melts where bubble nucleation is inhibited and bubble growth is limited. These conditions seem more likely to be attained during magma ascent between 400-100 MPa. If CO₂ degassing is diffusion-limited during magma ascent, which seems likely due to the presence of sparse and relatively small CO₂ bubbles, our (equilibrium) fractional degassing scenario at 20-100 MPa would result in the removal of too much CO₂ from the bulk magma. Diffusion-limited CO₂ degassing in the shallow cavity is also

possible, and is suggested by the transient decrease in H₂O/CO₂ throughout a fountaining period, which suggests that gas erupted later in the fountaining cycles have had more time for CO₂ to diffuse into them. However, Figure 2b shows that the H₂O/CO₂ ratio according to an open-system degassing scenario decreases sharply from >15 to ~1 to with increasing pressure above ~2 MPa, and these trends could additionally point to gas bubbles derived from a greater depth towards the end of fountaining.

We discuss the potential for diffusion-limited degassing to affect the composition of the gas erupted during fountaining in the revised version of the manuscript, by adding the following text to the manuscript on lines 115-118:

While diffusion-limited CO₂ degassing could also explain the elevated H₂O/CO₂ ratios observed during fountaining, the nucleation and storage of H₂O-rich gas bubbles in a shallow cavity would contribute to a closer approach to equilibrium³³.

33. Pichavant, M. et al. Generation of CO₂-rich melts during basalt magma ascent and degassing. *Contrib. to Mineral. Petrol.* 166, 545–561 (2013).

132: Is the lava plastered on the walls sufficient to produce the amount of gas required to achieve the observed flux at the required ratios?

In the revised manuscript, we explicitly states that the gas is derived from a combination of passive degassing of the outgassed lava within the lava lake as well as that plastered on the crater walls on lines (138-140):

At the start of the repose period (Fig. 3b), H₂O/CO₂ molecular ratio gradually increases and SO₂/HCl molecular ratio gradually decreases, as the magma filling the lava lake and plastered on the crater walls becomes progressively outgassed.

Note that our measurements do not allow us to delineate absolute gas fluxes occurring during fountaining or repose periods.

133: The decrease in H₂O/CO₂ 'as the lake fills' is inferred to occur before fountaining starts. How is the CO₂ rich gas released? Presumably this must be from partly stagnated magma that infilled conduit between shallow storage and vent. Are the ratios and fluxes observed consistent with this? If not, what is happening? Is the idea that the excess is actually vanguard gas from the shallow storage? This could matter for the physical interpretation.

In the revised version of the manuscript, we have enhanced discussion of transient changes in H₂O/CO₂ during repose and fountaining. The gas released during repose periods reflects passive degassing of the partly outgassed magma that refilled the lava lake after fountaining, as well as more deeply-derived (from the shallow cavity), CO₂-rich gas slugs that leak through the conduit penetrate the surface of the lava lake, which become more abundant before the onset of fountaining. This interpretation would be consistent with the measured ratios, as well as the inferred physical mechanism driving episodic fountaining. The relevant text (lines 141-143) now reads:

As the lava lake fills to close to its maximum level (Fig. 3c), the H₂O/CO₂ ratio decreases and SO₂/HCl increases, reflecting the CO₂- and SO₂-richer gas flowing through the conduit that equilibrated just beneath the surface.

176: 'shallow cavity at depth' is an awkward construction.

We have changed this to “shallow cavity”.

229: It's not clear to me which values of initial volatile concentrations were used in the calculations. The text gives values for 'primitive melt' and different values for 'enriched melt' but doesn't unambiguously state which were used for the D-Compress and SolEx calculations.

We state this more clearly now on lines 263-264:

We used the composition of the enriched melt for the magma degassing calculations shown in Figure 2.

References:

Witham, F. and Llewellyn, E.W., 2006. Stability of lava lakes. *Journal of Volcanology and Geothermal Research*, 158(3-4), pp.321-332.

Reviewer #2 (Remarks to the Author):

Review of near-surface magma flow instability.....

Summary

This is an excellent addition to the literature of lava fountaining eruption. The gas data was gathered at a temporal resolution equal or better than any previous spectroscopic study. The authors document for the first time (globally) a clear two-stage pattern of magma outgassing. The one area that needs to be enhanced to make the paper suitable for **Nature** is a much more powerful discussion/conclusion section. The data presented here must be used to evaluate critically the competing conceptual models for lava fountaining described in the early part of the text. To my mind the data presented here makes it clearer that one of two proposed models for lava fountaining, annular flow, is totally untenable. The authors should strengthen the paper by stating unequivocally, their position with respect to the merits of the two competing models for Fagradalsfjall.

In light of the comments from Reviewer 1, we have revised our conceptual model and strengthened discussion of the comparison between our proposed conceptual model and the collapsing foam (CF) and rise-speed-dependent (RSD) models. The relevant discussion is found on lines 179-190:

Our proposed conceptual model combines elements of both the rise speed dependent (RSD) and collapsing foam (CF) models for lava fountaining in the framework of pressure fluctuations acting on a lava lake-conduit-cavity system. Similar to the CF model, the build-up and release of gas accumulated at the top of a cavity acts as a trigger for the onset of fountaining. However, sustained fountaining does not reflect the near-instantaneous collapse of a foam layer and the establishment of annular flow conditions in the conduit, but rather progressive nucleation and coalescence of bubbles as magma depressurizes and ascends, as proposed by the RSD model. As the duration of a fountaining event depends on the balance between the supply of melt and gas through the shallow cavity and the rate of magma outgassing and drain-back into the conduit, fountaining is episodic rather than continuous because the rate at which melt and gas recharge the cavity is slower than the rate at which the cavity evacuates during fountaining.

General comments

1) I am confused reading the text from line 130 onwards and comparing it to the caption for Figure 3. Where is the 'shallow cavity' mentioned in the text visible on the cartoons on the left of Figure 3. The caption for cartoon B talks about the 'cavity' underlying the base of the conduit. Does this mean the cavity is the wide area at the bottom of each sketch and if so is it part of the reservoir? Or is the reservoir significantly deeper and not shown on the cartoons? It would help a lot if an approximate depth could be stated for this cavity.

We acknowledge that Figure 3 was crowded in the previous version of the manuscript. In the revised version of the manuscript, we have enlarged this sub-panel and included a line that points to the inferred cavity, labeled "shallow cavity". The cavity is indeed represented by wider area at the bottom of each of the four sub-panels, and is part of the shallow magma plumbing system; the deeper reservoir of magma (as represented by the dyke intrusion in sub-panel A) is not represented in subpanels B-F. In the new version of Figure 3, we have also included the approximate depth for the shallow cavity (~100 m). The revised version of Figure 3 is shown below:

Figure 2. Sketch of the lava fountaining cycle at Fagradalsfjall. **A.** Three-dimensional perspective of eruption site, with vent 5 (the locus of eruption activity of the time, yellow star) fed by a melt conduit connected to a dyke at ~1 km depth. The cross-sections in **B-E** focus in on an area where a lava lake is connected to magma upflow from depth that supplies melt (red) and gas (blue) through a conduit. **B.** The level of the lava lake decreases as pressure at the base of the conduit equilibrates with the pressure in the underlying cavity. **C.** Ascending gas bubbles from the cavity transport gas and melt to the surface, increasing the level of the lava lake. Increasing pressure from the deeper lake causes more gas to accumulate in the conduit. **D.** Once magmastic head at the top of the conduit is less than the pressure in the cavity, accumulated gas empties from the cavity into the conduit, leading to slug coalescence and intense fountaining, which feeds surface lava flows. As gas is evacuated from the conduit, melt refills the conduit via drain-back. **E.** Lower pressure at the base of the conduit promotes gas exsolution via magma decompression. The progressive outgassing of the magma in the reservoir causes a gradual waning in the intensity of the fountaining, until dense, outgassed melt drains back into the conduit. The cycle repeats as melt and gas are supplied from depth. The lower panel on the right shows the spectroscopic measurements for one of these fountaining cycles (see Fig. 1), relating **F.** H₂O/CO₂ and **G.** SO₂/HCl to the fountaining cycle.

2) In the cartoons labelled B and C. (Figure 3) there seems to be a lava flow perched above the cone on the right hand side, which is well above the level of lava in the vent and so the flow appears to be perched and decoupled from source. Is this intended?

This was intended, as the video taken during the measurement period on May 5 (uploaded with the Supplementary Information) showed that the surface lava flow was mainly fed during fountaining, and as the level of the lava lake dropped during repose the lava flow at the surface was decoupled from the source. However, this changed somewhat throughout the eruption. Other videos (<https://www.youtube.com/watch?v=QluD3JgfhXc&t=181s>) taken during the intermittent fountaining showed that the surface lava flows were still connected to the lava lake during repose, albeit there was very little outflow from the crater. In the revised version of Figure 3, we do not as strongly indicate the disconnect between the lava flow and the lava lake during repose, but rather stress that the lava flows are fed most strongly during fountaining.

What are the noteworthy results?

The key result is the recognition of changing gas chemistry with time during fountaining episodes, which reflects a progressive decrease in the depth at which magma outgassing is occurring.

We believe that this may have been a misinterpretation of our gas chemistry data. In the previous version of the manuscript, we did not stress variations during fountaining to changes in the depth of magma degassing, but simply showed that the composition of the gas erupted during fountaining was consistent with equilibrium at low pressure (~1-2 MPa). In the revised manuscript and in particular in the revised version of Figure 3, we highlight changes in gas chemistry during an individual fountaining event. We describe this conceptual model in detail on lines 137-157:

Our high-resolution gas chemistry data suggest that fountaining was driven by cyclical decompression of a shallow magma-filled cavity (Figure 3). At the start of the repose period (Fig. 3b), the $\text{H}_2\text{O}/\text{CO}_2$ ratio gradually increases while the SO_2/HCl ratio gradually decreases, as the magma filling the lava lake and plastered on the crater walls becomes progressively outgassed. As the lava lake fills to close to its maximum level (Fig. 3c), the $\text{H}_2\text{O}/\text{CO}_2$ ratio decreases and SO_2/HCl increases, reflecting the CO_2 - and SO_2 -richer gas flowing through the conduit that equilibrated just beneath the surface. As the conduit and lake become progressively more gas-rich, the average density of the magma filling the lake and conduit decreases, and the magmastatic head exerted on the top of the cavity decreases. Once the pressure at the base of the conduit is less than the pressure in the cavity, gas that has accumulated in the cavity during the repose period is released, triggering lava fountaining (Fig. 3d). Fountaining depressurizes the conduit and cavity and promotes further outgassing of melt in the underlying cavity, acting as positive feedback. As gas is sourced from progressively greater depths in the cavity throughout a fountaining event, and there is more time for slower-diffusing SO_2 and CO_2 to enter gas bubbles, the $\text{H}_2\text{O}/\text{CO}_2$ ratio decreases and SO_2/HCl increases during fountaining (Fig. 3c-d). Eventually, magma in the cavity becomes progressively outgassed, and the sharp decrease in the SO_2/HCl ratio records when the measured gas is no longer derived from the cavity but rather from already outgassed lava. Drain-back of melt from the crater into the conduit and inflow of melt and gas from the dyke into the shallow cavity recharge the system and the cycle repeats.

Thus, the changing gas chemistry during a fountaining event (in particular, $\text{H}_2\text{O}/\text{CO}_2$ gradually decreasing and SO_2/HCl gradually increasing) could suggest the opposite what was implied by the

reviewer, as this signature is consistent with a slightly higher pressure of equilibration. However, as we note in the manuscript, these trends could also result from diffusion of CO₂ into later-erupted gas bubbles.

Will the work be of significance to the field and related fields?

Yes, if the authors strengthen up the conclusion section. This should force models to critically examine all pre-existing models for the dynamics of magma scent accompanying lava fountaining. In addition, there have been few groups prepared to make volatile measurements of lava fountaining on fine temporal and spatial scales and this study should encourage others, particularly in the USA and Italy, to make similar measurements.

In the revised manuscript, we have strengthened the contextualization of the proposed conceptual model with pre-existing models for episodic lava fountaining (see response to comment above). Further, we have strengthened the conclusions section on lines 203-209:

Our gas chemistry data show that the much shorter period fluctuations observed at Fagradalsfjall reflects cyclical decompression of a shallow magma-filled cavity, in contrast to the episodic lava fountains at Etna or Kīlauea, for which much longer repose periods indicate control by deeper processes associated with upper crustal magma reservoirs. Moreover, our dataset highlights how measuring the composition of the erupted volatile phase on fine temporal and spatial scales can provide important insight into the physical and chemical processes governing the eruption dynamics.

Does the work support the conclusions and claims, or is additional evidence needed?

It is the reverse, there needs to be more forceful conclusions, based on these excellent data.

Are there any flaws in the data analysis, interpretation and conclusions? No.

Is the methodology sound? Does the work meet the expected standards in your field? Yes.

Is there enough detail provided in the methods for the work to be reproduced? Yes.

The manuscript is clearly written and well constructed. Figure 1 is excellent and is the heart of the paper. Figure 3 is very crowded and confusing particularly with respect to the 'cavity' and the 'reservoir'. It could do with some revision to increase the size of the key cross-sections.

We have revised Figure 3 in the updated version of the manuscript (see above), including labeling the 'cavity' (as described above), and increasing the size of the key cross-sections.

Minor Comments

These are made on the attached copy of the manuscript.

We have adapted most of the suggested revisions from the reviewer into the revised version of the manuscript, including changing cyclic/cyclical/intermittent to episodic.

Reviewer #3 (Remarks to the Author):

This is an interesting paper centered on an impressively cyclic gas chemistry dataset from the

Fagradalsfjall eruption of 2021.

The authors collected FTIR spectra during repeating lava fountaining events and found repeating patterns of gas chemistry and eruptive behavior. The authors then link the observational data to models of degassing behavior to elicit a model of the behavior driving the cyclic fountaining.

This is a nice dataset and I believe the paper will be a worthy contribution to **Nature** Communications following some edits and revisions.

My main concern currently is that while the authors outline the observed gas chemistry cycles and how they relate to modeled gas behavior and composition, they then describe three existing models for lava fountaining and identify one that fits with their observations, but fail to adequately discount the other two that they've already introduced. They don't include enough information about how they've drawn on their data to arrive at their conceptual model. In particular, given some similarities to their conceptual cartoon (Figure 3) and the 'foam collapse' model that is introduced but not mentioned again, I'd imagine that readers would want to know why the authors feel the 'pressure cycles' model is a better fit to the chemistry data than is the 'foam collapse' model. The discussion should explain how the other models would result in chemical results incongruous with their observations. As currently written, even if the conclusions about the best fitting model are correct, it feels as if the discussions and explanations of how those conclusions were reached are lacking.

I also feel that the manuscript would benefit from the removal of the use of the SolEx model. SolEx is only applicable to systems with an oxygen fugacity of greater than +0.5 units above the NNO buffer; this system is more than a log unit below that limit. The authors themselves mention doubts about SolEx given that it does not allow for inclusions of fractional degassing in the modeling. Ultimately chlorine behavior doesn't appear to bear on results and conclusions much; the authors (rightly) focus on the behavior of ratios involving H₂O, CO₂, and SO₂. Given its minor role in the paper and questions regarding the appropriateness of the model, perhaps SolEx should be omitted. There should be a discussion of/justification for still using SolEx, if it is to be kept in the paper.

We have omitted the SolEx degassing calculations from the new version of the manuscript.

The other main thing that would improve the clarity of the paper is expanded discussion of what is happening during the 'repose' periods. Particularly around lines 43-59, the text mentions 'repose' periods, but what is happening in the vent during that time? I don't think it's actually described here. Is there lesser fountaining out of sight (roiling)? A stagnant lava lake? A drained conduit passively releasing gas? Without that description, it's not clear what the 'repose' gas represents. Similarly, lines 58-59 mentions lava being the IR source during fountaining, but what is the IR source during repose? Overall, information regarding the repose measurements is lacking and needs to be included in a revised version.

In order to clarify the nature of the lava lake made during repose, we have added the following sentences on lines 58-59:

During repose, a roiling lava lake is present in the crater at a level beneath the crater rim. As a result, the IR source during repose is hot lava plastered on the walls of the crater.

Moreover, we have clarified the significance of the repose gas measurements on lines 107-110:

The higher H₂O/CO₂ and H₂O/SO₂ ratios and lower SO₂/HCl ratios measured during repose are characteristic of lava outgassing: the residual degassing of primarily H₂O and some HCl from lava in the lake and uppermost region of the conduit that has already lost most of its initial volatile load at the second stage of gas segregation and fountaining.

Other, smaller suggestions and issues are brought up for specific lines/items below.

I thank the authors and the editors for the invitation to review this paper.

- Lines 27/30 – Is there a reason for use of cyclic vs cyclical? If they are meant to have the same meaning, being consistent with one seems preferred.

Based on the comments from Reviewer 2, we have changed cyclic to episodic throughout the manuscript, with the exception of the title, which we have chosen to retain as cyclic.

- Line 41 – This should mention comparable melts, not magmas (magma encompasses the gas phase along with crystals and melt; in this case, the bulk magma is indeed different given that the shallow-sourced eruptions have already lost CO₂).

We have changed “magma” to “melt” in the line in question.

- Lines 64-69 – I believe readers would benefit from being able to see the mentioned gas ratios in table form, either in addition to or instead of the in-text comparisons of differences in values between repose and fountaining (and in addition to the running time series plots in Figure 1). Perhaps this could be added to Table 1.

We have added the mentioned gas ratios to Table 1.

- Figure 1 – Intensity decreases during the repose intervals (<10k, and even <5k in some cases). I understand that units are arbitrary, and may vary from instrument to instrument, but these values seem quite low (perhaps problematically so?). How does low intensity affect noise in your spectra and thus in the retrievals? Particularly of trace species like HF and CO? Can the authors quantify error or confidence levels for the retrievals, and does it differ between repose and fountaining? This would help confirm for readers that changes in gas chemistry are real rather than artifacts of possible noise in spectra with low intensity.

In the new version of Figure S5, we have included error bars showing the error of the fits.

Figure S5. Scatter plots of column amounts for pairs of gases to determine the ratios. Red circles represent measurements during fountaining and blue during repose. H₂O, CO₂, and CO have been corrected for atmospheric contamination. Errors associated with retrievals of column amounts from measured spectra shown with grey bars. Note that the errors for many of the repose measurements with low gas contents are less than the symbol size. A) H₂O vs. CO₂, B) H₂O vs. SO₂, C) SO₂ vs. HCl, D) CO₂ vs SO₂, E) HCl vs HF, F) CO₂ vs. CO.

As shown in Figure S5, the absolute errors are greater for the fountaining measurements are greater than the repose measurements due to the greater absolute gas contents during fountaining. While the relative errors can be higher during repose compared to fountaining, likely due to the increased noise in the lower intensity spectra, this is most evident for HF, for which the mean error for the HF retrievals is 14% during repose and 11% during fountaining. In contrast, the mean error for the CO retrievals is ~31% during both fountaining and repose. For the other gases, the mean errors associated with the retrievals are lower (H₂O ~5% during fountaining and repose, CO₂ ~10% during fountaining and ~8% during repose, SO₂ ~ 6% during fountaining and repose, HCl ~12% during fountaining and ~8% during repose). We believe that these levels of error are reflected by the uncertainty bounds (\pm two standard deviations) presented in Table 1 for the obtained gas ratios during fountaining and repose.

- Lines 80-83 – Can the authors add approximate equivalent depths for the pressures here, as they do in line 85?

Although there is some uncertainty in the depth range of degassing for these gas species, depending on the melt composition and volatile content, we have added equivalent depths to the pressures ranges in these lines. Lines 80-83 now read:

As is typical for water-poor basaltic magmas, CO₂ begins to exsolve at high pressures (~500 MPa, ~20 km depth), followed by SO₂ at intermediate pressure (~200 MPa, ~7 km depth), and H₂O and HCl at low pressures (<20 MPa, <1 km depth).

- Line 83 – Here the paper mentions the ratio as being by volume. Plots in Figures 1-3 are molar ratios. Yes, they work out to the same for gases, but best to be consistent with language and change in-text mentions to molar to match the figures.

We have changed “by volume” to “molar” in the new version of the manuscript.

- Line 93 – This has the figure letters as lowercase, but in the figures and captions, they are capital letters. Should they be the same? I’m not positive about journal style requirements.

The journal style requirements suggest the use of lowercase bold letters. All of the sub-panel labels in the figures and figure captions have been modified accordingly.

- Lines 107-109 – The H₂O and CO₂ being higher than predicted can make sense based on finite diffusion times, but it’s curious that there’s less S in the glass than predicted given that it’s a slow diffuser. Should this be explored more? Lerner et al 2021 showed S in Kīlauea 2018 matrix glasses more along the lines of your expected 700 ppm. I wonder about the discrepancy.

We agree that there is significant uncertainty in the sulfur degassing models. In the paper cited by the reviewer (Lerner et al., 2021), the authors report a wide range of measured S concentrations in matrix glasses (from <100 ppm to ~900 ppm), and link this variability to the extent of degassing during ascent, eruption, and surface flow. The reported value of ~186 ppm for Fagradalsfjall is consistent with a partially/highly degassed lava flow from Kīlauea. In addition, the basaltic melt investigated by Lerner et al. (2021) was sulfide-saturated, as demonstrated by the presence of sulfide globules. In contrast, the magma at Fagradalsfjall never reached sulfide saturation, and no sulfide globules were identified in tephra. The discrepancy between the measured and modeled residual S content could reflect that the Fagradalsfjall melts never reached sulfide saturation, whereas the model for sulfur solubility incorporated in D-Compress is based on experimental studies (e.g., Lesne et al. 2015) wherein melts were equilibrated with solid sulfide minerals (e.g., pyrrhotite). While this highlights the uncertainty of the S degassing models, particularly at low pressures, we focus on H₂O/CO₂ in our manuscript, and feel that a detailed discussion of sulfur degassing models (including disequilibrium effects) is beyond the scope of this paper.

Lerner, A. H. et al. The petrologic and degassing behavior of sulfur and other magmatic volatiles from the 2018 eruption of Kīlauea, Hawai‘i: melt concentrations, magma storage depths, and magma recycling. *Bulletin of Volcanology*, **83** (2021).

Lesne, P., Scaillet, B. & Pichavant, M. The solubility of sulfur in hydrous basaltic melts. *Chem. Geol.* 418, 104–116 (2015).

- Line 146 – This relates to the comment above about lines 43-59; this mention of a lava lake comes out of nowhere. Addressing the lack of information about the repose periods earlier in the paper will make this sentence/mention less surprising.

By adding the above discussion of the lava lake during repose on lines 58-60, we hope that the mention of the lava lake in this section is less surprising.

- Line 151 – The text cites other studies that arrive at this conduit diameter, but it may be worth explicitly (and briefly) mentioning here how that diameter was determined in those studies.

Eibl et al. (2023) do not estimate the conduit diameter from their seismic data, they just link an increase in seismic amplitudes over time to an increase in the conduit cross-sectional area. As far as we know, the acoustic study of Lamb et al. (2022) is the only one to directly estimate the conduit radius from geophysical data, which is compared with drone images recorded at around the same time. Eibl et al. do however propose the short conduit directly linked to a shallow cavity. To reduce the potential for confusion, we have moved the reference to Lamb et al. (2022) to the middle of the sentence directly after the '5-10-m diameter conduit' statement, and have added a statement regarding how the conduit diameter was estimated. Lines 160-165 in the revised manuscript read:

Although our data do not allow us to visualize the geometry of the shallow magma plumbing system, the lava lake was fed via by a 5–10-m-diameter conduit³⁸ likely connected to a larger cavity below the surface¹². The conduit diameter was estimated from modelling of acoustic signals from bursting gas slugs at the end of each lava fountain episode, and compared with drone images captured during a repose interval³⁶.

12. Eibl, E. P. S. *et al.* Evolving shallow conduit revealed by tremor and vent activity observations during episodic lava fountaining of the 2021 Geldingadalir eruption, Iceland. *Bull. Volcanol.* **85**, (2023).

38. Lamb, O. D. *et al.* Acoustic observations of lava fountain activity during the 2021 Fagradalsfjall eruption, Iceland. *Bull. Volcanol.* **84**, 1–18 (2022).

- Lines 169-171 – This is concluding a progressive decrease in source volatile content over time. It would be helpful if the authors could discuss/hypothesize about why this might be.

We have decided that our dataset is not on it's own enough support for a hypothesis linking the changing duration of fountaining/repose periods to a progressive decrease in source volatile content over time. In the revised manuscript, the relevant text (lines 190-198) reads:

While the fountaining periodicity was remarkably consistent throughout the measurement period on 5 May (Fig. 1), the duration of a cycle (including fountaining and repose) varied between 3 and 20 min throughout the six weeks of this cyclical behavior, with a trend towards lengthening period over time¹². Assuming the physical mechanism described by our conceptual model is applicable to the entire episodic fountaining stage, an increase in the repose period would suggest a progressive decrease in the rate of deep recharge of melt and gas, as more time would be required to generate the pressure fluctuations necessary to induce fountaining. However, other factors likely come into play, including changes in the conduit radius due to mechanical or thermal erosion¹².

12. Eibl, E. P. S. *et al.* Evolving shallow conduit revealed by tremor and vent activity observations during episodic lava fountaining of the 2021 Geldingadalir eruption, Iceland. *Bull. Volcanol.* **85**, (2023).

- Line 230 – The FTIR data was later than this tephra eruption date. Was there any documented change in erupted composition over the course of the eruption that might call for using a different starting composition?

Although the FTIR date is later than this tephra date, this was the closest available data point that we had to constrain melt composition. As documented in Halldórsson *et al.* (2022), there were significant changes in the erupted composition between March – April 2021, with more later erupted melts distinguished by higher MgO and K₂O/TiO₂. The changes in melt composition after the episodic fountaining stage were less pronounced. Therefore, we believe that there is likely a reasonable correspondence between the melt composition used for the degassing calculations and the composition of the erupted melt on May 5th.

Halldórsson, S. A. *et al.* Rapid shifting of a deep magmatic source at Fagradalsfjall volcano, Iceland. *Nature* **609**, 529–534 (2022).

- Line 257 and following section – How did the acoustic measurements compare to visual assessments of fountaining versus repose? Were the acoustic data necessary (and would they be for future studies) or could that have been done with the video that was being scrutinized for fountain heights already?

Figure 1 in the submitted manuscript as well as Figure 5 in Lamb *et al.* (2022) show that the acoustics closely track the fountain height measurements, but the acoustics remain high for a short amount of time after the fountain heights drop away. This indicates that there was still activity within the vent that the camera could not see, as the activity is blocked by the crater rim and outside of the field of view. For future studies, there is no guarantee one would be able to obtain a convenient perspective into the vent for cameras, so acoustic measurements would be vital for understanding when eruptive activity is happening.

Lamb, O. D. *et al.* Acoustic observations of lava fountain activity during the 2021 Fagradalsfjall eruption, Iceland. *Bull. Volcanol.* **84**, 1–18 (2022).

- Figure S5 – Even after correction, there seems to be somewhat of an issue with the fountain data for H₂O/SO₂. Possibly also in the H₂O/CO₂ plot. Either a curve at lower concentrations, or an offset from the origin (incomplete correction?) such that the data distribution doesn't fit well amidst the linear slopes. Is this real? A problem with the data? How has that affected the apparent slopes/bulk ratios (as in the plots of this figure)?

We thank the reviewer for this comment. We consider that this slight offset resulted from an underestimation in the background H₂O content. We assumed a background H₂O content of 3900 ppm, as calculated based on the meteorological data from the measurement period (relative humidity of 50%, pressure 985 hPa, temperature 3 °C). However, the meteorological data show slight variations throughout the measurement period. To reduce the issue pointed out by the reviewer, where the corrected H₂O measurements did not start at the origin, we increased the assumed background H₂O atmospheric content from 3900 ppm to 4150 ppm, replotted the recorrected H₂O data in Figure 1, and recalculated the gas compositions accordingly. This resulted in

slightly lower H₂O content of the gas during fountaining and repose, as well as a very slight decrease in the ratios H₂O/ CO₂ and H₂O/SO₂. All reported measurements in the revised manuscript reflect this slightly changed background contribution of H₂O to the measured spectra.

- Figure 3a – The purple star doesn't look purple and is impossible to recognize with the skewed perspective unless very zoomed in. I suggest changing the wording and the symbology for the current size/configuration of the figure.

We have changed the color of the star to yellow and increased the size of the sub-panel. We hope that Figure 3 is easier to interpret in the revised manuscript.

REVIEWER COMMENTS

Reviewer #3 (Remarks to the Author):

This paper leverages a very impressive gas chemistry dataset to infer the mechanisms of gas release leading to periodic lava fountaining. The manuscript is generally well-written and should be of great interest to those in the realms of eruption dynamics and volcanic gas chemistry, as well as the broader volcanology community as a whole. The paper is certainly worthy of publication, though some revisions are necessary. In a handful of places, particularly in the discussion section, there are statements or decisions/assumptions made by the authors that are not adequately explained/supported. Rewording and/or expansion of these sections would enhance clarity for readers. I also had some questions about Figures 2 and 3. Comments are outlined below.

Line 48 – It reads strangely to highlight only the fast diffusivity of water when the focus is also on CO₂. CO₂ is indeed a slower diffuser and could affect these results via disequilibrium degassing. Figure 2 – Lines 93 and 95 refer to Figure 2 as having panels b-e. There is no 'e'. Additionally in Figure 2a, where does the Cl curve come from? D-Compress does not include chlorine, so it needs to be explained how this was calculated/arrived at. Line 97 mentions a measured H₂O/CO₂ of 12 +/- 3, but that doesn't seem to match the gray bar in Figure 2b, which says it represents the measured ratios plus 2 SD. Maybe the +/- 3 isn't the same as your 2 SD, but the gray doesn't even appear to be centered on 12. (Based on Table 1, should the 12 be rounded to 13 from 12.6 anyway? This makes the gray box seem even more ill-placed.) It's also not entirely clear (to me, at least) where the pink/red range of inferred equilibrium pressures comes from – is it the equivalent depth of the cavity? Or is it based on the measured composition matching the open-system curve? Matching the closed-system 20MPa curve? More clarity is needed.

Lines 99-101 – Bubbles rising more rapidly than the melt seems counter to the minutes-long sustained fountaining you're measuring and discussing. It's my understanding that such behavior, in the RSD, tends more toward sporadic strombolian bursts, while bubbles that can't rise quickly enough to escape feed more sustained Hawaiian-style fountaining, which aligns more with closed-system degassing. Accordingly, I don't follow the logic here for favoring open-system degassing for the fountaining, especially when the post-CO₂-loss closed-system compositions also match the observations.

Lines 115-118 – Is the only reason you favor the H₂O-rich bubbles over diffusion-limited CO₂ degassing because it's closer to equilibrium? Is that a valid assumption for such a process as dynamic as lava fountaining?

Line 131-132 – I don't see how increasing P in the lava lake would increase both the size and the amount of gas slugs. According to the Witham model, it does lead to larger slugs, but fewer (and the rate of gas input to the system is constant). Also, I think the description of the process in lines 132-133 is overly simplistic. Yes, a large slug can trigger the downflow, but it's not the presence of a slug itself that does so, it's the pressure instability induced by the outgassing of such a slug. I would reword this to improve clarity.

Figure 3 – I think part A of this figure could be improved by presenting it as a simple 2-D cross-section rather than a 3-D perspective. With no external context, it's hard to see it as 3-D, which just leaves it looking oddly skewed and distorted (the yellow star doesn't look like a star). The lava flows are also already presented nicely in the supplemental figures, so it's not clear what they add to this figure, especially in the skewed 3-D view. In part B of the figure, there should be bubbles. There are always bubbles – however your model would indicate they're distributed for the scenario in B, they should be depicted in the cartoon. For parts D and E of the figure, could the lava fountain be portrayed better? Right now, there are no bubbles shown, and the solid red patch doesn't allow for distinguishing the dispersed fountain spray from the more contiguous lava feeding the flow.

Figure 3 caption – The portion of the caption referring to panel C says that increasing pressure from the deeper lake causes more gas to accumulate in the conduit. This wording isn't clear – do you mean that the increased lake P (owing to more volume in the lake) is inhibiting exsolution into bubbles, thus limiting the amount of bubbles rising and entering the lake? Or something else? This mention of increasing pressure in the lake also seems counter to part D of the caption, which relies on less pressure in the lake to induce the fountaining.

Line 140-142 – Is this essentially precursory leakage of some of the accumulated gas/foam?

Line 199-200 – This is semantics, but is this really an endmember on a spectrum if it essentially

incorporates both CF and RSD while also relying on pressure fluctuations? That seems more a middle-ground between the CF and RSD processes rather than an endmember. Adding in pressure fluctuations, and this is more the center of a ternary.

Line 298 – mention what RUV is. I believe a TV/media outlet, but it should be mentioned in the text. This also comes up in the supplementary material.

Line 301 – explain/cite the LUV colorspace.

Reviewer #4 (Remarks to the Author):

I have been asked to enter the review process following the withdrawal of a previous reviewer. The remarks to the author from the previous reviewer have been responded to and the manuscript updated. I provide, therefore, my review of the revised manuscript.

I attach an annotated pdf copy of the manuscript with more detailed feedback. Below I summarise my main review comments.

The synchronous gas chemistry and surface process data are very noteworthy, high quality results of great interest to a wide community. The manuscript is well crafted with excellent figures.

The results are used to develop a qualitative conceptual physical model of gas-liquid separation constrained by the gas chemistry, surface observations and the existing literature. I did find the model difficult to follow and make the following comments.

The cyclical process proposed seems complicated. Physically, the most straightforward model would be cyclical bubble raft growth at a roof and collapse into a conduit (Vergnolle & Jaupart experiments). This mechanism superimposes a cyclical gas flux onto a relatively constant magma flux at the vent.

A shallow (~ 100 m depth) 'chamber' is proposed in which gas can accumulate. Is there any reason why the dyke top could not be at this level? The dyke top would provide a greater accumulation volume without invoking a bubble raft thickness that seems a little implausible.

I also struggle with conduits from the dyke top at 1 km depth to 6 (or so) vents being stable. The dyke breaching the surface then focusing on 6 (or so) vents as effusion rate declines would be a typical process, but the dyke top would likely be shallower than 1 km.

The conceptual diagram (Figure 3) shows the foam collapse model, with the collapse feeding a Taylor bubble (gas slug) or 'bubble' of high gas mass fraction that ascends to the surface ~ 100 m above. The literature for the dynamics of this process has not been incorporated into the model. The dynamics are important because the dynamic overpressuring of the gas slug and development of explosive behaviour is all focussed in to the top few 10s of metres of the conduit.

Current thinking on the nature of lava fountains needs incorporating, i.e., neither explosive or effusive.

In summary, I think that the physical conceptual model is unduly complex (Occam's razor) but that this is an important paper that should be developed.

Steve Lane

Reviewer #3 (Remarks to the Author):

This paper leverages a very impressive gas chemistry dataset to infer the mechanisms of gas release leading to periodic lava fountaining. The manuscript is generally well-written and should be of great interest to those in the realms of eruption dynamics and volcanic gas chemistry, as well as the broader volcanology community as a whole. The paper is certainly worthy of publication, though some revisions are necessary. In a handful of places, particularly in the discussion section, there are statements or decisions/assumptions made by the authors that are not adequately explained/supported. Rewording and/or expansion of these sections would enhance clarity for readers. I also had some questions about Figures 2 and 3. Comments are outlined below.

Line 48 – It reads strangely to highlight only the fast diffusivity of water when the focus is also on CO₂. CO₂ is indeed a slower diffuser and could affect these results via disequilibrium degassing.

We have changed the emphasis of this sentence to stress the contrasting degassing behaviors of H₂O and CO₂ rather than the contrasting diffusive timescales (lines 47-50):

Given the pressure sensitivity of H₂O solubility in silicate melt as it approaches atmospheric pressure²⁰⁻²¹, and the contrasting behavior of CO₂, which exsolves at greater depth²², the measured H₂O/CO₂ ratio reflects the depth and pressure conditions of gas-melt equilibration.

In the revised manuscript, we also clearly note the potential role of the slow diffusivity of CO₂ in contributing to the elevated H₂O/CO₂ ratios on lines 109-111:

As gas bubble expansion is rapid in the near-surface, causing the dynamic overpressure that drives the fountaining³⁰, we therefore suggest that these measurements during fountaining preserve an approach to equilibrium attained at shallow depth. However, it should be noted that diffusion-limited CO₂ degassing could also contribute to elevated H₂O/CO₂ ratios observed during fountaining³¹.

Further discussion of the potential importance of diffusion-limited CO₂ degassing is provided in response to another question below.

20. Dixon J. E., Stoper E. M. and Holloway J. R. (1995) An Experimental Study of Water and Carbon Dioxide Solubilities in Mid-Ocean Ridge Basaltic Liquids. Part I: Calibration and Solubility Models. *J. Petrol.* **36**, 1607–1631.
21. Iacono-Marziano, G., Morizet, Y., Le Trong, E. & Gaillard, F. New experimental data and semi-empirical parameterization of H₂O–CO₂ solubility in mafic melts. *Geochim. Cosmochim. Acta* **97**, 1–23 (2012).
22. Métrich, N. & Wallace, P. J. Volatile abundances in basaltic magmas and their degassing paths tracked by melt inclusions. *Rev. Mineral. Geochemistry* **69**, 363–402 (2008).
30. Del Bello, E., Llewellyn, E. W., Taddeucci, J., Scarlato, P. & Lane, S. J. An analytical model for gas overpressure in slug-driven explosions: Insights into Strombolian volcanic eruptions. *J. Geophys. Res. Solid Earth* **117**, (2012).

31. Pichavant, M. et al. Generation of CO₂-rich melts during basalt magma ascent and degassing. *Contrib. to Mineral. Petrol.* 166, 545–561 (2013).

Figure 2 – Lines 93 and 95 refer to Figure 2 as having panels b-e. There is no 'e'. Additionally in Figure 2a, where does the Cl curve come from? D-Compress does not include chlorine, so it needs to be explained how this was calculated/arrived at. Line 97 mentions a measured H₂O/CO₂ of 12 +/- 3, but that doesn't seem to match the gray bar in Figure 2b, which says it represents the measured ratios plus 2 SD. Maybe the +/- 3 isn't the same as your 2 SD, but the gray doesn't even appear to be centered on 12. (Based on Table 1, should the 12 be rounded to 13 from 12.6 anyway? This makes the gray box seem even more ill-placed.) It's also not entirely clear (to me, at least) where the pink/red range of inferred equilibrium pressures comes from – is it the equivalent depth of the cavity? Or is it based on the measured composition matching the open-system curve? Matching the closed-system 20MPa curve? More clarity is needed.

Our sincere apologies for our oversight in the manuscript regarding Figure 2. Panel e, which in the first version of the manuscript showed the calculated SO₂/HCl ratio using SolEx to model HCl degassing, was removed from the text of resubmitted manuscript, but not removed from the figure caption or manuscript text.

Regarding the absolute values of the gas ratios, model calculations in the revised manuscript assumed a slightly different background H₂O correction as compared with the initially submitted manuscript. These recalculated ratios were not updated in the grey bar in the figure, which accounts for the offset between what was shown in figure 2 and indicated in Table 1 and the text.

In the revised manuscript, we have changed the location of the grey boxes to correspond to the mean plus 1 SD indicated in Table 1 as well as the text. These ratios are consistent throughout the text, figures, and table. Furthermore, we have removed the red box suggesting an equilibration pressure of 1-2 MPa. In the revised text, rather than suggesting this specific range for the gas equilibration pressure, we suggest that comparison of the model calculations and measured gas compositions strongly suggests near-surface gas equilibration pressure. This is consistent with both the geochemical modeling of magma degassing as well as the conceptual model for the mechanism driving the intermittent fountaining.

The updated version of figure 2 is provided below as well as the relevant text on lines 99-105:

Figures 2b-d show that the measured H₂O/CO₂, H₂O/SO₂ and CO₂/SO₂ ratios during fountaining (grey shading in Fig. 2) are all consistent with low pressure equilibration. In particular, the measured H₂O/CO₂ of 12±3 during fountaining suggests equilibration at near-atmospheric pressures for the closed-system degassing scenario, and pressure of ~2 MPa for an open-system degassing scenario. Although the magmatic depths corresponding to these pressure estimates depend on assumed magma density and vesicularity, these results suggest gas-melt equilibration at depths of <100 m, i.e., very near to the surface.

Figure 2. Degassing of the Fagradalsfjall melt. **A.** Dissolved CO_2 , H_2O , S and Cl concentrations in the melt as a function of pressure. The blue area shows the inferred pressure at which a first stage of gas loss occurs. **B.** $\text{H}_2\text{O}/\text{CO}_2$, and **C.** $\text{H}_2\text{O}/\text{SO}_2$, and **D.** CO_2/SO_2 . Closed-system degassing of initial melt with no gas loss shown with thin dashed line. Closed- and open-system degassing after gas loss at 20-100 MPa shown with solid and thick-dashed lines, respectively, labeled with the assumed pressure of gas loss. Measured ratios during fountaining (plus one standard deviation) shown with grey bars. Inferred pressure of gas equilibration shown in red. Inferred depths are shown on the secondary y-axes assuming a magma-density of 2600 kg m^{-3} (ref 29) and a vesicularity of 0%.

29. Murase, T. & McBirney, A. R. Properties of Some Common Igneous Rocks and Their Melts at High Temperatures. *GSA Bull.* **84**, 3563–3592 (1973).

Lines 99-101 – Bubbles rising more rapidly than the melt seems counter to the minutes-long sustained fountaining you’re measuring and discussing. It’s my understanding that such behavior, in the RSD, tends more toward sporadic strombolian bursts, while bubbles that can’t rise quickly enough to escape feed more sustained Hawaiian-style fountaining, which aligns more with closed-system degassing. Accordingly, I don’t follow the logic here for favoring open-system degassing for the fountaining, especially when the post- CO_2 -loss closed-system compositions also match the observations.

Although our models suggest that open-system behavior and CO_2 loss at depth is to explain the H_2O -rich gas compositions (see response to question below), we agree upon reconsideration that the system could still behave in a closed-system manner closer to the surface, due to the shallow depth of degassing and relative fast magma ascent rate resulting in coupled gas-melt flow. As described above, in the revised manuscript, we no longer favor open-system degassing, but simply note the correspondence between modeled and measured gas compositions suggesting equilibration at near-surface pressures. The relevant paragraph encompasses lines 99-108:

Figures 2b-d show that the measured $\text{H}_2\text{O}/\text{CO}_2$, $\text{H}_2\text{O}/\text{SO}_2$ and CO_2/SO_2 ratios during fountaining (grey shading in Fig. 2) are all consistent with low pressure equilibration. In particular, the measured $\text{H}_2\text{O}/\text{CO}_2$ of 12.6 ± 3 during fountaining suggests equilibration at near-atmospheric pressures for the closed-system degassing scenario, and pressure of $\sim 2 \text{ MPa}$ for an open-system

degassing scenario. Although the magmatic depths corresponding to these pressure estimates depend on assumed magma density and vesicularity, these results suggest gas-melt equilibration at depths of <100 m, i.e., very near to the surface. As gas bubble expansion is rapid in the near-surface, causing the dynamic overpressure that drives the fountaining³¹, we therefore suggest that these measurements during fountaining preserve an approach to equilibrium attained at shallow depth.

Lines 115-118 – Is the only reason you favor the H₂O-rich bubbles over diffusion-limited CO₂ degassing because it's closer to equilibrium? Is that a valid assumption for such a process as dynamic as lava fountaining?

The measured composition of the gas erupted during fountaining indicates H₂O molar abundance of ~90%. As shown in Figure 2 and discussed in the section regarding geochemical modeling of magma degassing, we suggest that this water-rich gas composition is the result of loss of deeply exsolved CO₂ at depth. This suggestion is based on: 1) the inferred initial relative abundance of H₂O and CO₂ in the basaltic melt (molar ratio close to ~1); 2) the contrasting pressure dependence of the solubilities of H₂O and CO₂, leading to CO₂ exsolution at depth and H₂O exsolution near the surface; and 3) the measured low concentrations of CO₂ in groundmass glass (18±3.9 ppm), which indicate an approach to equilibrium. If the elevated H₂O/CO₂ ratios measured in the erupted gas were produced solely by diffusion-limited CO₂ degassing, rather than open system CO₂ loss at depth, we would expect higher concentrations in the groundmass glass (Pichavant et al., 2013).

Thus, while we believe that loss of CO₂ at depth is the main process controlling the relatively low abundance of CO₂ in erupted gas, in the revised manuscript, we note that CO₂ degassing could be diffusion-limited (as described above and on lines 108-110):

However, it should be noted that diffusion-limited CO₂ degassing could also contribute to elevated H₂O/CO₂ ratios observed during fountaining³¹.

Moreover, in our revised manuscript, we acknowledge the importance of diffusion-limited during two stages:

- 1) During residual degassing of the magma at the surface, lines 111-119 and lines 166-171:

The higher H₂O/CO₂ and H₂O/SO₂ ratios and lower SO₂/HCl ratios measured during repose are characteristic of outgassing³²⁻³³ of residual lava in the lake and uppermost region of the conduit that has already lost most of its initial volatile load during lava fountaining. The measured CO₂, H₂O, and S concentrations in groundmass glass are 18±3.9, 720±63, and 186±100 ppm, respectively¹⁸, whereas equilibrium degassing models predict residual CO₂, H₂O, and S concentrations in magma at atmospheric pressure of <1, ~400, and ~700 ppm, respectively (Fig. 3b,c). Thus, the models do not capture the last stages of magma decompression and degassing as pressure approaches atmospheric, and non-equilibrium (e.g., diffusional) effects likely come into play^{31,34}.

...

At the start of the repose period (Fig. 3b), the magma contained in the lake, conduit and plastered on the crater walls was erupted and degassed during a previous fountaining

episode. The relatively high H₂O/CO₂ ratio (~30-50) and low SO₂/HCl ratio (5-10) represents Rayleigh distillation of the crystallizing magma, with higher solubility volatile species such as H₂O and Cl remaining in the melt in contrast to SO₂ and CO₂, which are rapidly lost from the magma at the surface^{32,33}.

- 2) As a factor leading to decreasing H₂O/CO₂ ratio and increasing SO₂/HCl ratio during fountaining, (lines 186-195)

We suggest that the collapse of a gas-rich pocket at the roof of the cavity causes fountains to reach their maximum heights soon after the onset of fountaining, and that fountaining is sustained following the attainment of peak fountain heights because fountaining depressurizes the conduit and cavity and promotes further degassing of melt in the underlying cavity, acting as a positive feedback. As our model assumes that the rapidly ascending gas slugs preserves an approach to gas-melt equilibrium acquired at shallow depth, the decrease in the H₂O/CO₂ ratio from ~15 to ~8 (Fig. 3c) and increase in the SO₂/HCl ratio from ~25 to ~45 (Fig. 3d) during fountaining could represent gas slugs sourced from progressively greater depths in the cavity throughout a fountaining event. Alternatively, this compositional change could result from slower-diffusing SO₂ and CO₂ entering later-formed bubbles³⁴.

18. Halldórsson, S. A. *et al.* Rapid shifting of a deep magmatic source at Fagradalsfjall volcano, Iceland. *Nature* **609**, 529–534 (2022).
31. Pichavant, M. *et al.* Generation of CO₂-rich melts during basalt magma ascent and degassing. *Contrib. to Mineral. Petrol.* **166**, 545–561 (2013).
32. Burton, M., Allard, P., Murè, F. & Oppenheimer, C. FTIR remote sensing of fractional magma degassing at Mount Etna, Sicily. *Geol. Soc. Spec. Publ.* **213**, 281–293 (2003).
33. Sigmarsson, O., Moune, S. & Gauthier, P.-J. Fractional degassing of S, Cl and F from basalt magma in the Bárðarbunga rift zone, Iceland. *Bull. Volcanol.* **82**, 54 (2020).
34. Freda, C., Baker, D. R. & Scarlato, P. Sulfur diffusion in basaltic melts. *Geochim. Cosmochim. Acta* **69**, 5061–5069 (2005).

Line 131-132 – I don't see how increasing P in the lava lake would increase both the size and the amount of gas slugs. According to the Witham model, it does lead to larger slugs, but fewer (and the rate of gas input to the system is constant). Also, I think the description of the process in lines 132-133 is overly simplistic. Yes, a large slug can trigger the downflow, but it's not the presence of a slug itself that does so, it's the pressure instability induced by the outgassing of such a slug. I would reword this to improve clarity.

We have changed our description of the Witham model in the revised manuscript (lines 129-136):

Another proposed mechanism for episodic activity is tied to dynamic pressure instabilities in lava lake/conduit systems³⁷⁻³⁸. Ascending gas slugs drive magma from the conduit into the lake, gradually filling the lake with magma and increasing the pressure at the base of the conduit, thereby reducing the flow of melt and gas into the lake. When a large slug drains through the conduit, the resulting void refills with denser outgassed magma from the lake. If the pressure at the base of the conduit exceeds that of the source reservoir, the flow at the base of the conduit reverses resulting in partial drainage of the lake.

37. Witham, F., Woods, A. W. & Gladstone, C. An analogue experimental model of depth fluctuations in lava lakes. *Bull. Volcanol.* **69**, 51–56 (2006).
38. Witham, F. & Llewellyn, E. W. Stability of lava lakes. *J. Volcanol. Geotherm. Res.* **158**, 321–332 (2006).

Figure 3 – I think part A of this figure could be improved by presenting it as a simple 2-D cross-section rather than a 3-D perspective. With no external context, it's hard to see it as 3-D, which just leaves it looking oddly skewed and distorted (the yellow star doesn't look like a star). The lava flows are also already presented nicely in the supplemental figures, so it's not clear what they add to this figure, especially in the skewed 3-D view. In part B of the figure, there should be bubbles. There are always bubbles – however your model would indicate they're distributed for the scenario in B, they should be depicted in the cartoon. For parts D and E of the figure, could the lava fountain be portrayed better? Right now, there are no bubbles shown, and the solid red patch doesn't allow for distinguishing the dispersed fountain spray from the more contiguous lava feeding the flow.

We have extensively revised Figure 3, replacing the 3D perspective with a 2D cross-section including measured topography along a profile corresponding to the strike of the dyke. We have also more clearly highlighted the role of bubbles and slug ascent in the cavity. We have also changed the way the lava fountain is represented to a more simplified “explosive” form, additionally highlighting the change in the height of the lava fountain throughout the fountaining cycle. The new version of Figure 3 is shown below.

Figure 1. Sketch of the lava fountaining cycle at Fagradalsfjall. **A.** Three-dimensional perspective of eruption site, with vent 5 (the locus of eruption activity at the time, yellow star) fed by a melt conduit connected to a partially-solidified dyke at ~ 1.5 km depth¹⁷. Topography⁵² along the axis of the dyke shown for pre-eruption conditions, 3 May 2021 and 21 Sep 2021. The cross-sections in **B-E** focus in on an area where a lava lake is supplied by melt (red) and gas bubbles ('slugs', white) through a conduit. **B.** At the beginning of repose, the crater is filled outgassed magma, and the level of the lava lake decreases as pressure at the base of the conduit equilibrates with the pressure in the underlying cavity. **C.** Supply of gas and melt from depth into the cavity causes ascending gas bubbles from the cavity to transport gas and melt to the surface, decreasing the average density of the magma in the conduit/lake and increasing the level of the lava lake. Gas bubbles accumulate on the roof of the cavity to form a foam. **D.** Ascent of a large slug through the conduit destabilizes the foam layer, which empties from the cavity into the conduit, leading to slug coalescence and tall fountains (~ 100 m). **E.** Lower pressures at the base of the conduit during fountaining promote gas exsolution in the shallow cavity via magma decompression. The progressive degassing of magma in the shallow cavity drives shorter fountains (~ 30 m) with a gradual waning in the intensity of the fountaining until dense, outgassed melt drains back into the conduit. The cycle repeats as melt and gas are supplied from depth. The lower panels show the spectroscopic

measurements for one of these fountaining cycles (see Fig. 1), relating F. H₂O/CO₂ and G. SO₂/HCl to the fountaining cycle.

Figure 3 caption – The portion of the caption referring to panel C says that increasing pressure from the deeper lake causes more gas to accumulate in the conduit. This wording isn't clear – do you mean that the increased lake P (owing to more volume in the lake) is inhibiting exsolution into bubbles, thus limiting the amount of bubbles rising and entering the lake? Or something else? This mention of increasing pressure in the lake also seems counter to part D of the caption, which relies on less pressure in the lake to induce the fountaining.

As shown above, we have reworded the caption for Figure 3 to reflect our updated model and the text in question has been removed.

Line 140-142 – Is this essentially precursory leakage of some of the accumulated gas/foam?

Yes. This is clarified in the revised manuscript on in the above caption for Figure 3 as well as lines 172-175:

As gas and melt are supplied from depth into the shallow cavity, gas accumulates to form a foam on the roof of the cavity, and some bubbles leak through the conduit, decreasing the average density of the magma filling the conduit and causing the upper surface of the lava to increase to maintain pressure equilibrium with a constant pressure source (Fig. 3c).

Line 199-200 – This is semantics, but is this really an endmember on a spectrum if it essentially incorporates both CF and RSD while also relying on pressure fluctuations? That seems more a middle-ground between the CF and RSD processes rather than an endmember. Adding in pressure fluctuations, and this is more the center of a ternary.

We agree with this interpretation and have revised the relevant section accordingly on lines 215-223:

Our conceptual model combines elements of both the rise-speed dependent (RSD) and collapsing foam (CF) models for lava fountaining in the framework of pressure fluctuations acting on a lava lake-conduit-cavity system. Similar to the RSD model, Hawaiian style lava-fountaining results due to the relatively rapid ascent rate of the magma, which allows coupled gas-melt flow to the near surface. Similar to the CF model, the fountaining periodicity reflects the dynamics of the collapse and growth of a foam layer at the roof of a cavity. Due to the shallow depth of the cavity, which permits H₂O degassing and gas-melt separation within, the pressure drop resulting from slug ascent can trigger collapse of this layer and the onset of fountaining.

Line 298 – mention what RUV is. I believe a TV/media outlet, but it should be mentioned in the text. This also comes up in the supplementary material.

This is clarified in the revised manuscript on lines 345-346:

Fountain heights were determined from video recordings. Movie S1 shows a time-lapse (64x) video of RÚV (Ríkisútvarpið, Iceland's national public-service broadcasting organization) camera footage during the FTIR monitoring period on 5 May 2021.

Line 301 – explain/cite the LUV colorspace.

In the revised manuscript we have added a citation⁵⁴ describing the CIE LUV colorspace.

54. Commission Internationale de l'Éclairage (CIE) (1986) Colorimetry. 2nd Edition, Publication CIE No. 15.2. Commission Internationale de l'Éclairage, Vienna.

Reviewer #4 (Remarks to the Author):

I have been asked to enter the review process following the withdrawal of a previous reviewer. The remarks to the author from the previous reviewer have been responded to and the manuscript updated. I provide, therefore, my review of the revised manuscript.

I attach an annotated pdf copy of the manuscript with more detailed feedback. Below I summarise my main review comments.

Many thanks for your detailed comments on the manuscript; we have considered each of the detailed points provided on the annotated pdf during preparation of our revised manuscript.

The synchronous gas chemistry and surface process data are very noteworthy, high quality results of great interest to a wide community. The manuscript is well crafted with excellent figures.

The results are used to develop a qualitative conceptual physical model of gas-liquid separation constrained by the gas chemistry, surface observations and the existing literature. I did find the model difficult to follow and make the following comments.

The cyclical process proposed seems complicated. Physically, the most straightforward model would be cyclical bubble raft growth at a roof and collapse into a conduit (Vergnolle & Jaupart experiments). This mechanism superimposes a cyclical gas flux onto a relatively constant magma flux at the vent.

In the revised manuscript, we have extensively revised and simplified the conceptual model describing the cyclical fountaining, drawing on observational evidence including lava fountain heights and seismo-acoustic measurements. The relevant text (lines 137-202) is provided below:

In addition to the obvious regularity of the fountaining cycles on the timescales of days-weeks, observations of the active vent in early May 2021 at Fagradalsfjall (vent 5) indicate that the level and intensity of roiling (degassing) in the “lava lake” in the upward-flaring conduit/vent system gradually increased in the final ~30-60 seconds of repose (Supplementary Materials, Movie S1). Fountain heights would then increase up to the maximum height (~100-130 m) over ~20 seconds, prior to an extended period (~3-4 minutes) of sustained low level fountaining

fluctuating at 20–50 m. These fluctuations in fountain height and intensity were mirrored by a transition from more acoustic-tremor-dominated seismic energy during the initial fountaining stage to more Strombolian-style activity with distinct high-amplitude impulsive waveforms¹⁰. The transition to repose would often be gradual, occurring over ~60 seconds with short bursts of activity at the end of fountaining, and then gentle spattering of the magma would persist throughout repose, with many small bubbles (~10 cm diameter) and occasional larger bubbles up to ~1 m in diameter. During repose, the upper surface of the lava “lake” often revealed a conduit that narrowed with depth; sometimes the magma was withdrawn from view¹¹. The conduit diameter was estimated to be 5–10 m based on modelling of acoustic signals from bursting gas slugs at the end of each lava fountain episode and drone images captured during repose¹⁰.

In light of these observations as well as our measurements suggesting that the gas erupted during fountaining equilibrated at near-surface pressures, we propose that fountaining was initiated by gas accumulated in a shallow, magma-filled cavity during the repose period. Collapse of a foam layer into the conduit could be triggered by dynamic pressure instabilities induced by slug ascent, as experiments have shown that pressures in the wake of slugs ascending through flared tubes decrease by 10^3 - 10^5 Pa due to the falling liquid film on the walls of the conduit^{39,40}. While slight pressure changes are unlikely to trigger fountaining if the conduit is connected to a deeper cavity (>0.5 km), in shallow cavities this becomes feasible due to the pressure sensitivity of H₂O degassing at low pressures, which can lead to accumulation of a H₂O-rich foam layer at the roof of a shallow cavity (Fig. 2b). While the size and amount of slugs will be greatest soon after the collapse of this foam layer, the lower density of the magma-gas mixture in the conduit during fountaining will depressurize the magma in the cavity, leading to inflow of fresh magma from depth as well as further H₂O degassing. This decompression-driven magma degassing represents the more extended period of lava fountaining following the attainment of peak fountain heights.

Our conceptual model uniting the physical mechanism driving the episodic lava fountaining with the measured gas chemistry at Fagradalsfjall is sketched in Figure 3. At the start of the repose period (Fig. 3b), the magma contained in the lake, conduit and plastered on the crater walls was erupted and partially degassed during a previous fountaining episode. The relatively high H₂O/CO₂ ratio (~30-50) and low SO₂/HCl ratio (5-10) represents Rayleigh distillation of the crystallizing magma, with higher solubility volatile species such as H₂O and Cl remaining in the melt in contrast to SO₂ and CO₂, which are rapidly lost from the magma at the surface^{32,33}. Melt-gas separation in the shallow cavity causes the formation of a foam on the roof of the cavity. During gas accumulation, some bubbles leak through the conduit, decreasing the average density of the magma filling the conduit and causing the upper surface of the lava to increase to maintain pressure equilibrium with a constant pressure source (Fig. 3c). In response to attainment of a critical foam thickness and the ascent of a large slug through the conduit, pressure at the base of the conduit falls below the pressure in the cavity, destabilizing the foam layer and triggering lava fountaining (Fig. 3d). Gas erupted during fountaining retains the approach to melt-gas equilibrium acquired in the shallow cavity, with lower H₂O/CO₂ ratio (~10-15) and higher SO₂/HCl ratio (~30-40). As the foam raft drains through the conduit, the magma in the cavity is depressurized and degassed, resulting in ascending slugs that drive more sporadic and shorter fountains. The sharp decrease in the SO₂/HCl ratio records when the measured gas is

no longer derived from magma degassing in the cavity but rather from already outgassed lava erupted at the surface. Drain-back of outgassed melt from the crater into the conduit and inflow of “fresh” melt from the dyke into the shallow cavity recharge the system and the cycle repeats.

We suggest that foam collapse causes fountains to reach their maximum heights soon after the onset of fountaining, and that fountaining is sustained following the attainment of peak fountain heights because fountaining depressurizes the conduit and cavity and promotes further degassing of melt in the underlying cavity, acting as a positive feedback. As our model assumes that the rapidly ascending gas slugs preserve an approach to gas-melt equilibrium acquired at shallow depth, the decrease in the H₂O/CO₂ ratio from ~15 to ~8 (Fig. 3c) and increase in the SO₂/HCl ratio from ~25 to ~45 (Fig. 3d) during fountaining could represent gas slugs sourced from progressively greater depths in the cavity throughout a fountaining event. Alternatively, this compositional change could result from slower-diffusing SO₂ and CO₂ entering later-formed bubbles³⁴. At the end of fountaining, the sporadic nature of the spattering indicates progressive outgassing of the magma, which results in more Strombolian-type eruptive behavior.

This updated conceptual model has been reflected in Figure 3 (shown below). Although our conceptual model incorporates aspects of the foam collapse model, with the addition of dynamic pressure instabilities induced by slug ascent as a trigger for foam collapse, it also provides an explanation for the more extended period of lava fountaining following the attainment of peak fountain heights, linking this to magma decompression induced by fountaining. We believe that this updated conceptual model honors both the gas chemistry data, as well as observations of fountain heights and the transition from acoustic-tremor-dominated seismic energy during the initial fountaining stage to more Strombolian-style activity with distinct high-amplitude impulsive waveforms.

10. Lamb, O. D. *et al.* Acoustic observations of lava fountain activity during the 2021 Fagradalsfjall eruption, Iceland. *Bull. Volcanol.* **84**, 1–18 (2022).
11. Eibl, E. P. S. *et al.* Evolving shallow conduit revealed by tremor and vent activity observations during episodic lava fountaining of the 2021 Geldingadalir eruption, Iceland. *Bull. Volcanol.* **85**, (2023).
32. Burton, M., Allard, P., Murè, F. & Oppenheimer, C. FTIR remote sensing of fractional magma degassing at Mount Etna, Sicily. *Geol. Soc. Spec. Publ.* **213**, 281–293 (2003).
33. Sigmarsson, O., Moune, S. & Gauthier, P.-J. Fractional degassing of S, Cl and F from basalt magma in the Bárðarbunga rift zone, Iceland. *Bull. Volcanol.* **82**, 54 (2020).
34. Freda, C., Baker, D. R. & Scarlato, P. Sulfur diffusion in basaltic melts. *Geochim. Cosmochim. Acta* **69**, 5061–5069 (2005).
39. James, M. R., Lane, S. J., Chouet, B. & Gilbert, J. S. Pressure changes associated with the ascent and bursting of gas slugs in liquid-filled vertical and inclined conduits. *J. Volcanol. Geotherm. Res.* **129**, 61–82 (2004).
40. James, M. R., Lane, S. J. & Chouet, B. A. Gas slug ascent through changes in conduit diameter: Laboratory insights into a volcano-seismic source process in low-viscosity magmas. *J. Geophys. Res. Solid Earth* **111**, 1–25 (2006).

Figure 2. Sketch of the lava fountaining cycle at Fagradalsfjall. **A.** Three-dimensional perspective of eruption site, with vent 5 (the locus of eruption activity at the time, yellow star) fed by a melt conduit connected to a partially-solidified dyke at ~ 1.5 km depth¹⁷. Topography⁵¹ along the axis of the dyke shown for pre-eruption conditions, 3 May 2021 and 21 Sep 2021. The cross-sections in **B-E** focus in on an area where a conduit. **B.** At the beginning of repose, the crater is filled with outgassed lava, and the level of the lava “lake” decreases as pressure at the base of the conduit equilibrates with the pressure in the underlying cavity. **C.** Melt-gas separation in the cavity causes bubbles accumulate on the roof of the cavity to form a foam. Some gas bubbles leak through the conduit to the surface, decreasing the average density of the magma in the conduit/lake and increasing the level of the lava lake. **D.** Ascent of a large slug through the conduit destabilizes the foam layer, which empties from the cavity into the conduit, leading to slug coalescence and tall fountains (~ 100 m). **E.** Lower pressures at the base of the conduit

during fountaining promote gas exsolution in the shallow cavity and influx of volatile-rich melt from depth. The progressive degassing of magma in the shallow cavity drives shorter fountains (~30 m) with a gradual waning in the intensity of the fountaining until dense, outgassed melt drains back into the conduit. The lower panels show the spectroscopic measurements for one of these fountaining cycles (see Fig. 1), relating **F.** H₂O/CO₂ and **G.** SO₂/HCl to the fountaining cycle.

A shallow (~ 100 m depth) 'chamber' is proposed in which gas can accumulate. Is there any reason why the dyke top could not be at this level? The dyke top would provide a greater accumulation volume without invoking a bubble raft thickness that seems a little implausible.

The estimated depth to the top of the dyke is based on deformational signals associated with the dyke intrusion (Sigmundsson et al., 2022). This citation has been added to the caption to Figure 3. However, to emphasize the poor constraints on the geometry of the shallow subsurface magma plumbing system, a question mark has been added to panel A in Figure 3 where the partially molten section of the dyke is connected to the shallow cavity via the proposed feeder dyke.

Sigmundsson, F. *et al.* Deformation and seismicity decline before the 2021 Fagradalsfjall eruption. *Nature* **609**, 523–528 (2022).

I also struggle with conduits from the dyke top at 1 km depth to 6 (or so) vents being stable. The dyke breaching the surface then focusing on 6 (or so) vents as effusion rate declines would be a typical process, but the dyke top would likely be shallower than 1 km.

Similar to the answer to the above question, the depth to the top of the dyke has been estimated based on geodetic studies (Sigmundsson et al., 2022). Due to the poor constraints on the geometry of the shallow magma plumbing system, and in particular the relationship between the top of the dyke and the depth of the shallow cavity, a question mark has been added to panel A in Figure 3, reflecting this uncertainty.

The conceptual diagram (Figure 3) shows the foam collapse model, with the collapse feeding a Taylor bubble (gas slug) or 'bubble' of high gas mass fraction that ascends to the surface ~ 100 m above. The literature for the dynamics of this process has not been incorporated into the model. The dynamics are important because the dynamic overpressuring of the gas slug and development of explosive behaviour is all focussed in to the top few 10s of metres of the conduit.

As described above, we have extensively revised as well as the text regarding the physical mechanism of fountaining. The focus of this study is linking measurements of gas chemistry to the physical mechanism generating intermittent fountaining, rather than the physics of gas ascent in the near surface. Therefore, we consider a detailed quantitative analysis of the physics of this process beyond the scope of this manuscript. However, in the revised manuscript, we also provide a citation that supports our conclusion that the rapid ascent and overpressuring of gas slugs would preserve conditions acquired in the shallow cavity (lines 105-108):

As gas bubble expansion is rapid in the near-surface, causing the dynamic overpressure that drives the fountaining³⁰, we therefore suggest that these measurements during fountaining preserve an approach to equilibrium attained at shallow depth.

30. Del Bello, E., Llewelin, E. W., Taddeucci, J., Scarlato, P. & Lane, S. J. An analytical model for gas overpressure in slug-driven explosions: Insights into Strombolian volcanic eruptions. *J. Geophys. Res. Solid Earth* **117**, (2012).

Current thinking on the nature of lava fountains needs incorporating, i.e., neither explosive or effusive.

In the revised manuscript, this is clearly stated in the introduction (lines 21-25):

Distinct from 'explosive' and 'effusive' eruption styles, which reflect magma fragmentation at depth in the conduit or at the surface, respectively, lava fountaining involves surface fragmentation under a condition of choked-flow, a behavior typical of rapidly ascending, low-viscosity, volatile-poor (<1 wt % H₂O) basaltic magmas³⁻⁷.

In summary, I think that the physical conceptual model is unduly complex (Occam's razor) but that this is an important paper that should be developed.

In the revised manuscript, we propose a conceptual model that is simpler than the previously submitted manuscript (see above), but additionally accounts for observations such as fountain height and acoustic measurements. There are some outstanding questions about when the different processes become dominant - and how these processes are affected by chamber shape/depth/size, the quasi-steady flow rate, the melt viscosity and gas content, gas slip speed, and the lake shape/cross-section with depth. These will need some further modelling and experiments to clarify. In the revised manuscript, fundamentally what we seek to convey is that to first order this is still an extraordinarily cyclic phenomenon with an essentially nondestructive mechanism (notwithstanding the lava overflow during fountaining and shifting period over time).

Thanks for your constructive feedback.

Steve Lane